# Scalable Kronecker-Factored Fisher Approximation for Neural Network Parameter Sensitivity

**Viktoriia Chekalina** [1 2]  **Daniil Moskovskiy** [3 4]  **Tatyana Matveeva** [5]  **Andrey Kuznetsov** [1 6]  **Evgeny Frolov** [3 5]

## Abstract

The Fisher Information Matrix (FIM) provides a principled geometric framework for parameter sensitivity in neural networks, but directly computing and using the full FIM is infeasible in high-dimensional models. As a result, most existing methods rely on diagonal approximations that discard important correlation structure. We introduce Matrix-free Fisher Factorization (MFF), a GPU-tractable algorithm that captures both diagonal and off-diagonal dependencies without materializing the full matrix. For post-training neural network layer compression, we prove that under Matrix-Variate Normal assumptions, MFF yields GFWSVD, a unique closed-form linear layer decomposition that optimally minimizes the expected second-order loss increase. Experiments on controlled numerical benchmarks with large neural networks show that GFWSVD achieves up to 50% compression while matching or exceeding state-of-the-art diagonal and activation-based baselines across most tasks, and it reliably avoids collapse in dense architectures such as Llama 3. Moreover, when used to initialize existing optimization pipelines (e.g., Dobi-SVD), GFWSVD better preserves accuracy at 40% parameter reduction in regimes where standard methods substantially degrade. Together, these results position MFF and GFWSVD as foundational algorithmic primitives for scalable, second-order-aware neural network approximation and parameter sensitivity.

## 1. Introduction

The Fisher Information Matrix (FIM) (Fisher, 1922) is widely used in neural networks to improve model efficiency, particularly during training and inference. However, computing and utilizing the full Fisher matrix is computationally infeasible for deep neural networks. To address this challenge, most existing approaches rely on simplified approximations – most commonly assuming a diagonal Fisher matrix (Frankle & Carbin, 2019; Wu et al., 2024; Soen & Sun, 2024; Moskovskiy et al., 2025). While computationally efficient, this assumption ignores important cross-correlations between model parameters. Motivated by the importance of accurately parametrising the FIM and of preserving correlation structure in LLMs, we propose a GPU-tractable Kronecker-based factorization algorithm, Matrix-free Fisher Factorization (MFF). MFF avoids explicit storage of the full Fisher matrix and reduces computation to operations on objects whose dimensions match those of individual linear layers.

To demonstrate its practical relevance, we prove that under the assumption of a Matrix-Variate Normal (MVN) distribution, the resulting factors yield an optimal low-rank compression of the weights. We validate this result empirically by proposing the Generalized Fisher-Weighted SVD (GFWSVD), a low-rank compression method derived from this formulation. GFWSVD consistently outperforms both diagonal Fisher approximations (FWSVD (Hsu et al., 2022)) and activation-based baselines (ASVD (Yuan et al., 2023), SVD-LLM (Wang et al., 2025c)). Specifically, GFWSVD achieves state-of-the-art results in up to 50% model compression with performance superior to existing methods. When used as a drop-in initialization for fine-tuning pipelines (Wang et al., 2025b), GFWSVD enables reliable fine-tuning at up to 40% parameter reduction.

Our main contributions are summarized as follows:

1. We derive a Matrix-free Fisher Factorization algorithm tractable on GPU for the Kronecker-structured FIM.

2. We demonstrate that accounting for Kronecker factors yields a provably optimal post-training neural network parameter sensitivity, instantiated as a low-rank layer compression under the MVN assumption.

3. We propose the GFWSVD compression method, which consistently outperforms diagonal and activation-based

---

[1] Fusion Brain [2] MSU [3] AXXX [4] Applied AI Institute [5] HSE University [6] Innopolis University. Correspondence to: Viktoriia Chekalina <sayankotor1@gmail.com>.

*Proceedings of the $43^{rd}$ International Conference on Machine Learning*, Seoul, South Korea. PMLR 306, 2026. Copyright 2026 by the author(s).

baselines across a range of compression regimes and serves as an effective initialization for existing fine-tuning pipelines.

**Conflict of Interest Disclosure.**  The authors declare no financial conflicts of interest related to this work. The models evaluated in this paper (BERT, Llama 2, and Llama 3.1) are publicly available and were not developed by the authors' affiliated organizations.

## 2. Related Work

It is important to note that our work sits at the intersection of second-order optimization, matrix factorization, and efficient model compression. We categorize related approaches based on how they approximate curvature and how they utilize it for parameter reduction.

The Fisher Information Matrix encapsulates the local geometry of the loss landscape, serving as a foundation for natural gradient optimization (Martens & Grosse, 2015) and continual learning (Kirkpatrick et al., 2017). To mitigate the quadratic scaling of the FIM, methods typically employ structural approximations. Diagonal approximations (Kirkpatrick et al., 2017; Wu et al., 2024) assume parameter independence and are widely used in pruning (e.g., SparseGPT (Frantar & Alistarh, 2023)). Kronecker-factored approximations (K-FAC) (Martens & Grosse, 2015; Grosse & Martens, 2016; Schnaus et al., 2021) capture richer correlations by approximating the curvature of linear layers as a one-term Kronecker product of inputs and gradients. Closely related, Koroko et al. (2022) extend this to a two-term Kronecker approximation. However, both K-FAC and Koroko et al. (2022) are designed for optimization, relying on dynamic moving averages during training. Applying this structure to *post-training* decomposition on a fixed dataset is computationally expensive ($\mathcal{O}(m^2 n^2)$), a bottleneck our MFF algorithm resolves via efficient matrix-vector products.

**Post-Training Low-Rank Decomposition.**  Standard SVD minimizes the Frobenius norm of the reconstruction error, treating all parameters uniformly. Recent works attempt to inject importance weighting into this process. Activation-based methods like ASVD (Yuan et al., 2023) and SVD-LLM (Wang et al., 2025c) scale weights based on input feature norms. While effective, activations serve only as a proxy for task sensitivity. Fisher Information-based methods (e.g., FWSVD (Hsu et al., 2022), FWTTM (Pletenev et al., 2023a;b)) explicitly incorporate loss curvature but are limited to the diagonal FIM approximation. Cross-layer methods like Basis Sharing (Wang et al., 2025a) exploit redundancy between layers rather than within them. Our proposed GFWSVD generalizes the intra-layer approaches: under the MVN assumption, the decomposition allows for a

closed-form solution that captures both row and column correlations, dominating diagonal and activation-based heuristics in theoretical expressivity.

Beyond standalone decomposition, low-rank approximation often serves as a component in complex pipelines. Methods like Dobi-SVD (Wang et al., 2025b) and BLAST (Lee et al., 2024) use SVD factors as an initialization for subsequent gradient-based training. Approaches like BitStack (Wang et al., 2024) combine iterative decomposition with 1-bit quantization of sign matrices to manage variable memory constraints. Similarly, second-order quantization methods like YAQA (Tseng et al., 2025) use Hessian information to optimize rounding. Our work focuses on the foundational linear algebra primitive – the optimal weighted decomposition. As such, it is orthogonal to pipeline-based methods and can serve as a theoretically grounded initialization (as we demonstrate with Dobi-SVD) or a drop-in replacement for the SVD steps within hybrid frameworks.

## 3. Background and Problem Formulation

In this section, we establish the connection between Fisher information over matrix variables drawn from Matrix-Variate Normal distribution and our approach to approximating the Fisher matrix via a Kronecker product decomposition. We then leverage this decomposition to develop an improved parameter sensitivity algorithm based on the Fisher-weighted SVD formulation.

### 3.1. Parameter Sensitivity in Layer Compression Tasks

Consider post-training weight compression (PTC) as an example of parameter sensitivity application. The compression perturbs model parameters $\theta \in \mathbb{R}^d$ and drives the deviation of the model's loss function $\mathcal{L}(\theta)$ in the proximity of an optimal point $\theta^\star$. Sensitivity to such perturbation can be naturally captured by the second-order expansion of the loss determined by the quadratic term involving the Hessian $\mathbf{H} = \mathbf{H}(\theta^\star)$ of the problem:

$$\Delta\mathcal{L} = \mathcal{L}(\theta) - \mathcal{L}(\theta^*) \approx \frac{1}{2}(\theta - \theta^*)^\top \mathbf{H}(\theta - \theta^*) \quad (1)$$

Compression optimization thus corresponds to minimizing the deviation $\Delta\mathcal{L}$ with respect to a compression $\theta = \mathcal{C}(\theta^\star)$ while considering the structured curvature encoded in $\mathbf{H}$:

$$\min_{\mathcal{C}} \; (\theta^\star - \mathcal{C}(\theta^\star))^\top \mathbf{H}(\theta^\star - \mathcal{C}(\theta^\star)), \quad (2)$$

where the optimization task is considered over a functional family of compression methods $\mathcal{C}$.

In real-world settings, working directly with $\mathbf{H}$ is often intractable due to its size and complex structure. Thus, solving the task in Eq. 2 requires finding good enough approximations of the Hessian that ideally capture its most important

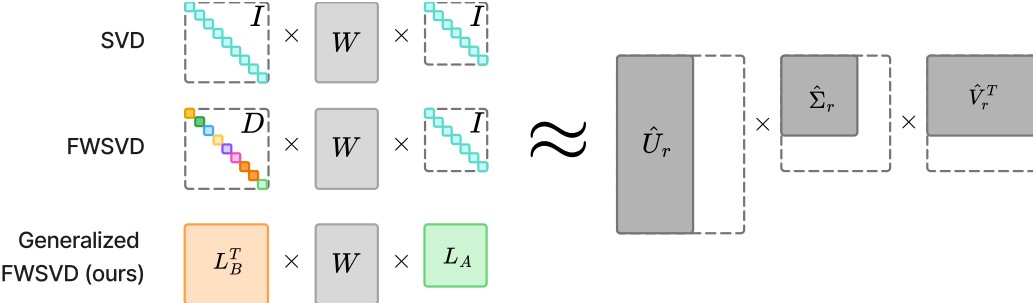

*Figure 1.* Generalization of the Weighted SVD frameworks. For standard SVD, the transformation matrices are identity matrices. For FWSVD, the left matrix is diagonal but not identity, and the right matrix is identity. For GFWSVD, both matrices are non-diagonal.

properties. As we show next, there is a certain class of approximations that align particularly well with this task.

### 3.2. MVN Distribution and Fisher Information

MVN distribution (Gupta & Nagar, 2018) extends the classical multivariate normal distribution to matrix-valued random variables, providing a structured approach to modeling dependencies within matrix rows and columns. Formally, a matrix $\mathbf{X} \in \mathbb{R}^{n \times m}$ follows an MVN distribution if its entries exhibit Gaussian properties with covariance structured across both dimensions. Its distribution is defined as

$$\mathbf{X} \sim \mathcal{MN}(\mathbf{M}, \boldsymbol{\Sigma}_1, \boldsymbol{\Sigma}_2), \tag{3}$$

where $\mathbf{M} \in \mathbb{R}^{n \times m}$ is the mean matrix, $\boldsymbol{\Sigma}_1 \in \mathbb{R}^{n \times n}$ and $\boldsymbol{\Sigma}_2 \in \mathbb{R}^{m \times m}$ are SPD matrices that capture correlations between rows and columns, respectively. The (non-degenerate) covariance is expressed as a Kronecker product $\boldsymbol{\Sigma}_2 \otimes \boldsymbol{\Sigma}_1$.

The log-probability density function of $\mathbf{X}$ has the form:

$$\log p(\mathbf{X}) \propto -\frac{1}{2} \operatorname{tr}\left( \boldsymbol{\Sigma}_1^{-1} (\mathbf{X} - \mathbf{M}) \boldsymbol{\Sigma}_2^{-1} (\mathbf{X} - \mathbf{M})^{\top} \right) \tag{4}$$

The corresponding Maximum Likelihood Estimation (MLE) task leads to minimization of trace in Eq. 4, which yields the Generalized Least Squares Matrix Decomposition objective (Allen et al., 2014). Subject to a rank-$r$ constraint, it reads:

$$\min_{\operatorname{rank}(\mathbf{X}) \leq r} \left\| \boldsymbol{\Sigma}_1^{-\frac{1}{2}} (\mathbf{X} - \mathbf{M}) \boldsymbol{\Sigma}_2^{-\frac{1}{2}} \right\|_{\mathrm{F}}^2. \tag{5}$$

The corresponding closed-form solution is obtained by means of standard SVD (Abdi, 2006):

$$\mathbf{X} = \boldsymbol{\Sigma}_1^{\frac{1}{2}} \hat{\mathbf{U}} \hat{\mathbf{S}} \hat{\mathbf{V}}^{\top} \boldsymbol{\Sigma}_2^{\frac{1}{2}}, \tag{6}$$

where $\hat{\mathbf{U}}\hat{\mathbf{S}}\hat{\mathbf{V}}^{\top} = \mathrm{SVD}_r(\boldsymbol{\Sigma}_1^{-\frac{1}{2}} \mathbf{M} \boldsymbol{\Sigma}_2^{-\frac{1}{2}})$ is the best rank-$r$ approximation. We note that the result also holds when matrix square roots are replaced with the corresponding Cholesky factors, which are typically easier to find.

Importantly, the MVN likelihood function (4) *admits an efficient closed-form representation of the second-order dependencies via the inverse Kronecker-factored covariance*: its Hessian $\mathbf{H} = \boldsymbol{\Sigma}_2^{-1} \otimes \boldsymbol{\Sigma}_1^{-1}$ coincides with the FIM at $\mathbf{M}$.

We additionally note that, by the formulation of the PTC task, model weights are assumed to be near the optimum, where the empirical Fisher serves as a practical approximation of the Hessian – this approach is widely used in model pruning (Singh & Alistarh, 2020). Nevertheless, the proposed method remains fully compatible with the true Fisher when gradients are replaced by pseudo-gradients.

### 3.3. Optimal Sensitivity with Layer Factorization

More generally, under regular conditions, i.e., a smooth twice-differentiable loss for MLE, Fisher Information also equals the expected local curvature (Hessian) at the optimum (Martens, 2020). This connection allows us to reframe the optimal compression problem (2) in terms of Fisher Information. We formalize this observation in Theorem 3.1.

**Theorem 3.1.** *Let* $\mathbf{W}^{\star} \in \mathbb{R}^{n \times m}$ *represent the optimal parameter weights matrix of a single layer of a trained neural network. Suppose that the following conditions hold.*

1. *The model is trained on a supervised task with an MLE objective (e.g., cross-entropy loss).*

2. *The empirical FIM restricted to the layer weights* $\mathbf{W}^{\star}$ *admits a Kronecker factorization* $\mathcal{I}_F \approx \mathbf{A} \otimes \mathbf{B}$.

3. *The weights* $\mathbf{W}$ *are drawn from the MVN distribution* $\mathcal{MN}(\mathbf{W}^{\star}, \mathbf{B}^{-1}, \mathbf{A}^{-1})$ *in the vicinity of optimum.*

*Under these conditions, an optimal rank-$r$ approximation of* $\mathbf{W}^{\star}$ *minimizing the expected loss increase* (1) *is given by:*

$$\widehat{\mathbf{W}}_r = \mathbf{L}_{\mathbf{B}}^{-\top} \widetilde{\mathbf{W}}_r \mathbf{L}_{\mathbf{A}}^{-1}, \tag{7}$$

*where* $\mathbf{L}_{\mathbf{A}}$ *and* $\mathbf{L}_{\mathbf{B}}$ *are obtained from the corresponding Cholesky decompositions* $\mathbf{A} = \mathbf{L}_{\mathbf{A}}\mathbf{L}_{\mathbf{A}}^{\top}$ *and* $\mathbf{B} = \mathbf{L}_{\mathbf{B}}\mathbf{L}_{\mathbf{B}}^{\top}$; $\widetilde{\mathbf{W}}_r$ *is obtained with the rank-$r$ truncated SVD of an auxiliary matrix* $\widetilde{\mathbf{W}} = \mathbf{L}_{\mathbf{B}}^{\top}\mathbf{W}^{\star}\mathbf{L}_{\mathbf{A}}$.

The first condition in Theorem 3.1 ensures the Hessian at convergence coincides with FIM, the second condition poses a structural assumption on its geometry, while the last one sets a convenient probabilistic model that ties the Kronecker-structured Fisher Information to the MVN likelihood (4). The formal proof of this theorem is provided in Appendix B.

Theorem 3.1 states that, under the MVN setting and Kronecker-structured FIM assumption, the optimum of problem (2) yields an SVD decomposition of the layer, weighted by the inverse square roots of the empirical Fisher Information's factor matrices. We elaborate on the practical feasibility and limitations arising from these theoretical assumptions in Appendix A. The procedure for obtaining this decomposition and the corresponding practical algorithm are elaborated in Sections 4 and 5.

## 4. Matrix-free Fisher Factorization

Suppose that we have a linear layer of a trained neural network with a weight matrix $\mathbf{W}$. Define $\mathbf{G}_i \in \mathbb{R}^{n \times m}$ as a matrix of weight gradients of $\mathcal{L}(\theta)|_{\theta=\text{vec}(\mathbf{W})}$ on the $i$-th batch, and $g_i = \text{vec}(\mathbf{G}_i) \in \mathbb{R}^{n \cdot m}$ – its flattened version. Then, Fisher Information $\mathcal{I}_F(\theta)$ can be defined as an empirical mean over all batches in a dataset $D$:

$$\mathcal{I}_F(\theta^\star) = \mathbb{E}\left[gg^\top\right] = \frac{1}{|D|} \sum_{i=1}^{|D|} g_i g_i^\top. \tag{8}$$

Kronecker product approximation is obtained by solving the corresponding minimization problem:

$$\min_{\mathbf{A},\mathbf{B}} \|\mathcal{I}_F - \mathbf{A} \otimes \mathbf{B}\|_\text{F}^2, \tag{9}$$

which is equivalent to finding the best rank-1 approximation of an $\mathcal{R}$-permuted Fisher matrix $\tilde{\mathcal{I}}_F = \mathcal{R}\mathcal{I}_F \in \mathbb{R}^{m^2 \times n^2}$, as established by Van Loan & Pitsianis (1993). Specifically, the singular vectors associated with the largest singular value of $\tilde{\mathcal{I}}_F$ yield the optimal factors $\mathbf{A}$ and $\mathbf{B}$. We summarize this efficient decomposition procedure in Algorithm 1.

---

**Algorithm 1** Matrix-free Fisher Factorization

---

**Require:** Gradients list $\{g_i\}_{i=1}^{|D|}$, $|D|$ – number of batches
1: $\mathcal{I}_F \leftarrow \frac{1}{|D|} \sum_{i=1}^{|D|} g_i g_i^\top$ (never materialized)
2: $\tilde{\mathcal{I}}_F \leftarrow \mathcal{R}\mathcal{I}_F = \frac{1}{|D|} \sum_{i=1}^{|D|} \mathbf{G}_i \otimes \mathbf{G}_i$ (never materialized)
3: $(u, \sigma, v^\top) \leftarrow$ leading singular triplet of $\tilde{\mathcal{I}}_F$ (Sec. 4.1)
4: $b \leftarrow u \cdot \sigma$
5: $a \leftarrow v$
6: $\mathbf{B} \leftarrow \text{reshape}(b, (m, m))$
7: $\mathbf{A} \leftarrow \text{reshape}(a, (n, n))$
8: **return** $(\mathbf{B}, \mathbf{A})$

---

### 4.1. Efficient Rank-1 Computation

The primary computational challenge of Algorithm 1 arises in performing SVD on the matrix $\tilde{\mathcal{I}}_F$.

Standard SVD is computationally intractable for large matrices, so we employ truncated SVD using the Lanczos method (Lanczos, 1950), which avoids explicit matrix construction and requires only the ability to multiply the matrix with a vector (matvec operation) from the left or right.

We show (see Appendix C) that permuted $\mathcal{I}_F$ for an $i$-th batch can be defined as the Kronecker product of the corresponding gradient matrices, yielding:

$$\tilde{\mathcal{I}}_F = \frac{1}{|D|} \sum_{i=1}^{|D|} \mathbf{G}_i \otimes \mathbf{G}_i. \tag{10}$$

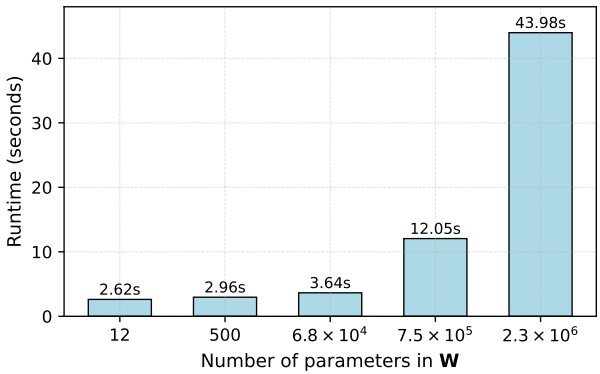

*Figure 2.* Empirical runtime for computing the Kronecker decomposition of the Fisher matrix for weight matrices of varying sizes.

Multiplying this matrix $\tilde{\mathcal{I}}_F$ by a vector $z$ on the right (given that $z = \text{vec}(\mathbf{Z}), \mathbf{Z} \in \mathbb{R}^{n \times n}$) yields:

$$\tilde{\mathcal{I}}_F z = \frac{1}{k} \left( \sum_{i=1}^{k} \mathbf{G}_i \otimes \mathbf{G}_i \right) \text{vec}(\mathbf{Z}). \tag{11}$$

Using the standard Kronecker product vectorization property $(\mathbf{K} \otimes \mathbf{L}) \text{vec}(\mathbf{C}) = \text{vec}(\mathbf{K}^\top \mathbf{C} \mathbf{L})$, we reduce the large matvec multiplication to a sequence of more compact matrix multiplications:

$$\tilde{\mathcal{I}}_F z = \frac{1}{|D|} \sum_{i=1}^{|D|} \text{vec}(\mathbf{G}_i^\top \mathbf{Z} \mathbf{G}_i) \tag{12}$$

that enable efficient iterative construction of the Krylov subspace required by the Lanczos procedure. The derivation for the right matvec is analogous (see Appendix D).

These operations enable the efficient approximation of the Fisher matrix for neural network layers at practical batch sizes. As stated in steps 1–2 of the Algorithm 1, the implementation is matrix-free: neither the full Fisher matrix $\mathcal{I}_F$ nor its permuted form $\tilde{\mathcal{I}}_F$ is ever explicitly formed. Instead, we downscale all computations to the layer size.

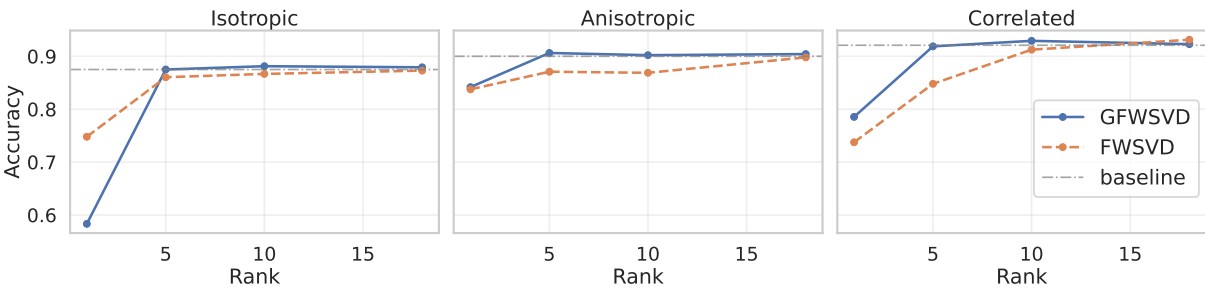

*Figure 3.* Comparison of diagonal and full Kronecker decompositions of a linear layer in an MLP.

## 4.2. Time Complexity

The time complexity of the truncated SVD computation for the matrix $\tilde{\mathcal{I}}_F \in \mathbb{R}^{m^2 \times n^2}$ consists of the cost of matvec multiplications and the orthogonalization step. In the naive case, $\tilde{\mathcal{I}}_F$ matvecs cost $\mathcal{O}\left(m^2 n^2\right)$, while orthogonalization cost is $\mathcal{O}((m^2 + n^2) \cdot r^2)$, where $r$ is the rank of the decomposition. With the structured approach from Eq. 12 that compactly implements matvec products via matrix multiplications, the overall complexity is reduced to $\mathcal{O}\left(mn^2 + m^2 n\right)$. The orthogonalization cost can be neglected due to $r \ll \min(m, n)$ in practical settings.

*Table 1.* Runtime for computing Kronecker factors of single linear layer on GPU.

| Model | Params in layer | Params in Hessian | Decomp. time (s) |
|---|---|---|---|
| BERT | $2.3\times10^6$ | $5.5\times10^{12}$ | 43 |
| Llama 2 7B | $45\times10^6$ | $2.0\times10^{15}$ | 183 |
| Llama 3.1 8B | $58\times10^6$ | $3.4\times10^{15}$ | 249 |
| Llama 2 13B | $70.8\times10^6$ | $4.9\times10^{15}$ | 313 |

Table 1 reports the empirical Hessian decomposition times for a single linear layer in LLMs of varying size, demonstrating that our accelerated algorithm is tractable even for large transformer models.

**Rank-1 Kronecker Fisher and multi-term extensions.** Rank-1 Kronecker approximation remains standard across various ML tasks (Martens & Grosse, 2015; Grosse & Martens, 2016; Schnaus et al., 2021). Figure 4 illustrates the spectral analysis of the $\mathcal{R}$-permuted Fisher matrix $\tilde{\mathcal{I}}_F$ in Llama-2-7B-chat to assess the feasibility of rank-1 approximation within Algorithm 1. The analysis indicates that, for some layers, the Kronecker spectrum decays slowly, so a rank-1 factor may leave a noticeable fraction of curvature unexplained.

## 5. Generalized Fisher-Weighted SVD

The post-training low-rank compression pipeline via GFWSVD is structured as follows:

1. **Model initialization.** A pre-trained model is provided, with parameters assumed to be at a (local) minimum of the loss function.

2. **Factor estimation.** Given a calibration dataset $\mathcal{D}$, the per-sample gradients are computed, and the corresponding Fisher factors are estimated according to Algorithm 1.

3. **Weight compression.** The closed-form solution (7) is applied to derive the optimally compressed model weights. The corresponding linear layer factorization reads

$$\mathbf{W}_1 = \hat{\mathbf{S}}^{\frac{1}{2}} \hat{\mathbf{V}}^\top \mathbf{L_A}^{-1}, \quad \mathbf{W}_2 = \mathbf{L_B}^{-\top} \hat{\mathbf{U}} \hat{\mathbf{S}}^{\frac{1}{2}},$$

where the $\hat{\mathbf{U}}, \hat{\mathbf{S}}, \hat{\mathbf{V}}^\top$ factors are obtained from the SVD of the auxiliary matrix $\mathbf{L_B}^\top \mathbf{W}^\star \mathbf{L_A}$.

**Singularity of Kronecker factors.** While effective, our method assumes an exact Kronecker factorization of the Fisher information matrix $\mathcal{I}_F$ into factors $\mathbf{A}$ and $\mathbf{B}$ (Eq. 9). This assumption may not hold in practice and can limit approximation quality and task sensitivity. In LLM experiments, we also observed cases where the estimated Kronecker factors were singular, requiring regularization (e.g., $\mathbf{A} \leftarrow \mathbf{A} + \alpha \operatorname{diag} \mathbf{A}$) to ensure positive definiteness and numerical stability. This regularization introduces only a small computational overhead of $\approx 5$ seconds per layer.

### 5.1. Relationship to Prior Works

We show that FWSVD (Hsu et al., 2022) is a special case of our generalized framework in Appendix E. The connection between our generalized approach, the classical SVD and FWSVD is depicted in Figure 1. Weighted SVD approaches can be interpreted as transforming the decomposed object – here, the weight matrix – into a new space where the low-rank approximation better aligns with the target task. Under this view, vanilla SVD corresponds to using identity transformations; FWSVD applies a diagonal (but non-identity) transformation on one side while keeping the other side as identity. In contrast, our method employs full, non-diagonal transformations on both sides, capturing richer structure in the parameter space.

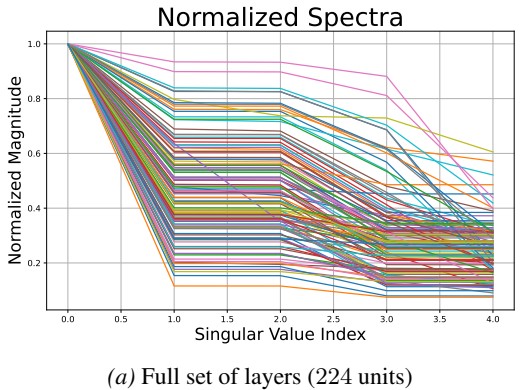

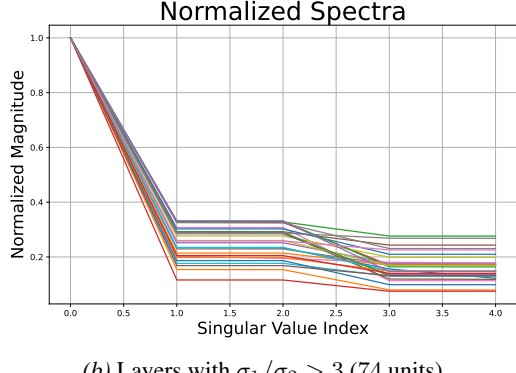

*(a)* Full set of layers (224 units)  *(b)* Layers with $\sigma_1/\sigma_2 > 3$ (74 units)

*Figure 4.* Singular value spectra of Llama-2-7B-chat linear layers.

## 6. Numerical Experiments

To validate our theoretical contributions, we conduct extensive numerical experiments. First, using a small model on synthetic datasets, we demonstrate the importance of explicitly accounting for correlations.

Second, we substantiate the practical usability of GFWSVD by experimenting with several Transformer architectures, including the encoder-only BERT model (Devlin et al., 2019) and the recent open-weight decoder-only LLMs Llama 2 (Touvron et al., 2023) and Llama 3.1 (Team, 2024). Our goal is to demonstrate the practical benefits of GFWSVD for post-training low-rank compression under standard fine-tuning and evaluation protocols. In all Transformer compression experiments we denote compression ratio as $1 - \frac{\text{compressed model}}{\text{original model}}$ .

### 6.1. Accounting for Correlations in Synthetic Datasets

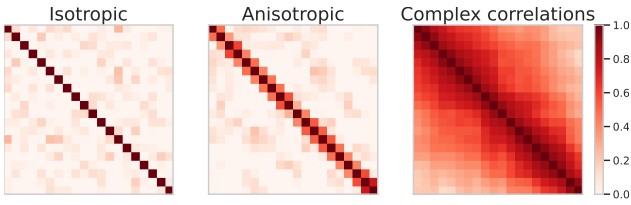

*Figure 5.* Heatmaps of the types of correlation matrices used for synthetic datasets.

To isolate the contribution of off-diagonal FIM elements, we evaluate our method in a controlled setting where the correlation structure of the input features is explicitly defined. We train MLPs from scratch on binary classification tasks generated from three distinct covariance regimes (visualized in Figure 5). In the *isotropic* case we use identity covariance ($\Sigma = I$), serving as a baseline with no correlations. In the *anisotropic* case we generate $\Sigma$ via a fixed randomized linear transform, introducing unstructured dependencies. In the *Toeplitz* matrix case we use structured covariance matrix

($\Sigma_{ij} = \rho^{|i-j|}$), simulating the dense, spatially decaying correlations often found in sequential data.

We decompose the target layers using our proposed rank-1 Kronecker estimation (Algorithm 1) and compare validation accuracy against a diagonal Fisher approximation and a baseline model without the decomposition. Figure 3 reports the validation accuracy as a function of rank $r$. The results confirm our theoretical intuition: while diagonal methods perform adequately in the *isotropic* regime, the GFWSVD factorization provides a substantial advantage as correlations increase. Notably, the performance gap is most pronounced in the *Toeplitz* setting, demonstrating that our method successfully captures structured off-diagonal dependencies that diagonal approximations discard.

We provide additional experimental results highlighting the importance of accounting for correlations in Transformer architectures in Appendix H (see Table 10).

### 6.2. Compressing the Transformer Encoder

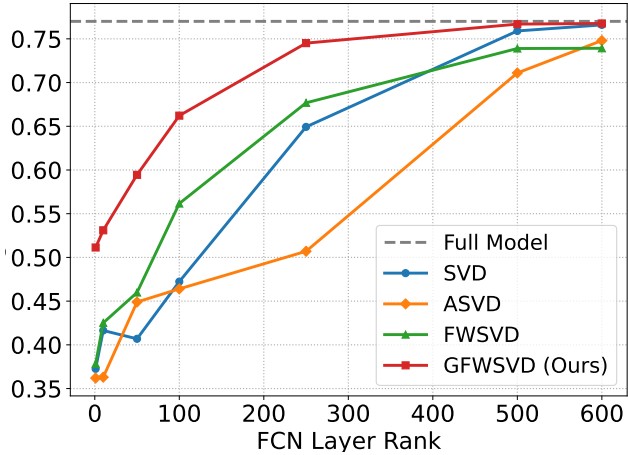

*Figure 6.* Macro-averaged GLUE performance of BERT model for different compression ranks.

*Table 2.* Macro-averaged GLUE performance of BERT for different compression ranks. Best results for each rank are in **bold**.

| Method / Rank | 600 | 500 | 250 | 100 | 50 | 10 | 1 |
|---|---|---|---|---|---|---|---|
| Compression ratio | 1% | 8% | 23% | 33% | 34% | 39% | 40% |
| SVD | **0.77** | 0.76 | 0.65 | 0.47 | 0.41 | 0.42 | 0.37 |
| ASVD | 0.75 | 0.71 | 0.51 | 0.46 | 0.45 | 0.36 | 0.36 |
| FWSVD | 0.74 | 0.74 | 0.68 | 0.56 | 0.46 | 0.43 | 0.38 |
| GFWSVD (Ours) | **0.77** | **0.77** | **0.75** | **0.66** | **0.59** | **0.53** | **0.51** |

In our experiments, we follow the *fine-tune then compress* pipeline, similar to FWSVD (Hsu et al., 2022). We begin by fine-tuning a pre-trained checkpoint[1] of BERT on a specific downstream GLUE task. Optimal fine-tuning hyperparameters (e.g., learning rate, batch size) are selected for each task using the Optuna framework (Akiba et al., 2019). During this stage, we also collect gradients to construct the FIM $\mathcal{I}_F$ and compute its Kronecker decomposition as described in Section 4.

Using the resulting Cholesky factors $\mathbf{L_A}$ and $\mathbf{L_B}$, we uniformly compress the fully connected layers of BERT by factorizing them into two smaller layers, following the method detailed in Section 3.1. The chosen layer-wise ranks and the resulting overall compression rate of the model are summarized in Table 2. We reproduce the ASVD method using the original authors code. For FWSVD, we incorporate the newly constructed FIM into the compression process.

We show the comparison results in Table 2 and Figure 6, with extended results in Appendix G in Table 9. On most of the GLUE tasks and considered compression ranks, our GFWSVD approach consistently outperforms both FWSVD and SVD, with particularly strong gains at lower ranks. While ASVD exhibits relatively poor performance on several tasks (QQP, QNLI), it occasionally surpasses GFWSVD – notably on SST2 under aggressive compression.

### 6.3. Compressing the Transformer Decoder

We evaluate our approach on the decoder-only models Llama 2 7B[2] and Llama 3.1 8B[3]. Since GFWSVD is purely analytical – containing no stochastic steps – we benchmark it against several competitive baselines of the same class of methods: diagonal FI-based low-rank approximation method FWSVD (Hsu et al., 2022), two activation-based methods – ASVD (Yuan et al., 2023) and SVD-LLM (Wang et al., 2025c), and per-layer relation-aware Basis Sharing (Wang et al., 2025a). Notably, ASVD and SVD-LLM both rely on activation-based weighting to gauge parameter importance, while Basis Sharing relies on correlations across layers in the entire model. In contrast, FWSVD and

our GFWSVD rely solely on gradient information, treating each layer as independent.

We measure perplexity on WikiText 2 (Merity et al., 2017) and PTB (Marcus et al., 1993) datasets, 5-shot reasoning performance on the MMLU benchmark (Hendrycks et al., 2021) and 0-shot performance on OpenBookQA (Banerjee et al., 2019), WinoGrande (Sakaguchi et al., 2021), HellaSwag (Zellers et al., 2019), PIQA (Bisk et al., 2020), ARC-E and ARC-C (Clark et al., 2018). Following prior works on low-rank approximation of LLMs (Wang et al., 2025c; Yuan et al., 2023), we test several compression setups, removing from 5% to 50% of original parameters.

Similar to previous works (Wang et al., 2025c; Yuan et al., 2023), we use a random sampled set of 9,600 samples as calibration data to generate gradients for further obtaining the factor matrices. For calibration data, we choose the FineWeb dataset (Penedo et al., 2024) due to its high quality and diversity. We collect gradients in a mini-batch manner with batch size of 128 with 75 total batches, and Lanczos iterations are performed over these batches. Gradients obtained in this manner are then used to compute $\mathbf{L_A}$ and $\mathbf{L_B}$, as well as the data needed for FWSVD. We emphasize that the batch-averaging operation does not commute with the matrix-vector product. Since $\tilde{\mathcal{I}}_F$ is an average of per-batch terms (Eq. 10), each Lanczos matvec is computed by applying every per-batch operator individually and only then averaging the results, rather than averaging the gradients first.

As in LLMs, uniform layer compression can disproportionately degrade performance by over-compressing critical layers and under-utilizing redundancy in less sensitive ones, so it is essential for each method to use a compression configuration that accounts for layer sensitivity. For both ASVD and SVD-LLM, we used the corresponding code released by the authors and re-ran the necessary compression pipelines for our checkpoint with all hyperparameters set to default values. For our approach, we adopted the method of per-layer importance scores as described in the ASVD work.

Tables 3 and 5 show that on Llama 2 7B and Llama 3.1 8B, and for compression levels up to 50%, GFWSVD substantially outperforms methods that decompose layers independently, both diagonal FWSVD and activation-aware ASVD. If we compare GFWSVD, which relies on parameter correlations within a layer, with Basis Sharing, which captures correlations across layers, GFWSVD outperforms Basis Sharing on Llama 3.1 8B at all compression levels and surpasses it on Llama 2 7B at 20% and 40% compression. This difference likely stems from stronger inter-layer correlations in the instruction-tuned Llama 3.1 8B model. We also observe that GFWSVD and Basis Sharing behave differently across tasks while maintaining consistent trends across compression ratios. For example, on PIQA, GFWSVD often

---

[1] `hf.co/google-bert/bert-base-uncased`
[2] `hf.co/meta-llama/Llama-2-7b-chat-hf`
[3] `hf.co/meta-llama/Llama-3.1-8B-Instruct`

*Table 3.* Performance of the Llama 3.1 8B Instruct compressed by various methods under compression ratios from 20% to 50% on WikiText-2 (WT2), PTB, and six common sense reasoning datasets. Lower is better for perplexity (↓), higher is better for accuracy (↑).

| METHOD | WT2↓ | PTB↓ | C. Ratio | ARC-C↑ | ARC-E↑ | HellaSwag↑ | PIQA↑ | WinoG.↑ | OpenBook↑ | AVG↑ |
|---|---|---|---|---|---|---|---|---|---|---|
| Full model | **7.20** | **11.50** | 0% | **0.52** ± 0.01 | **0.81** ± 0.01 | **0.59** ± 0.01 | **0.79** ± 0.01 | **0.73** ± 0.01 | **0.35** ± 0.02 | **0.63** |
| FWSVD | 354 | 864 | | 0.21 ± 0.01 | 0.38 ± 0.01 | 0.20 ± 0.01 | 0.60 ± 0.01 | 0.52 ± 0.01 | 0.17 ± 0.02 | 0.35 |
| ASVD | 145 | 1672 | 20% | 0.21 ± 0.01 | 0.33 ± 0.01 | 0.27 ± 0.01 | 0.61 ± 0.01 | 0.54 ± 0.01 | 0.15 ± 0.02 | 0.35 |
| Basis Sharing | **18.54** | 90.05 | | 0.34 ± 0.01 | **0.68** ± 0.01 | 0.42 ± 0.01 | 0.70 ± 0.01 | **0.65** ± 0.01 | **0.35** ± 0.02 | 0.52 |
| GFWSVD (Ours) | 22.57 | **42.40** | | **0.35** ± 0.01 | **0.68** ± 0.01 | **0.45** ± 0.01 | **0.75** ± 0.01 | 0.63 ± 0.01 | 0.33 ± 0.02 | **0.53** |
| FWSVD | 4372 | 6824 | | 0.21 ± 0.01 | 0.30 ± 0.01 | 0.26 ± 0.01 | 0.57 ± 0.01 | 0.51 ± 0.01 | 0.14 ± 0.02 | 0.33 |
| ASVD | 1456 | 4232 | 30% | 0.22 ± 0.01 | 0.30 ± 0.01 | 0.25 ± 0.01 | 0.58 ± 0.01 | 0.52 ± 0.01 | 0.16 ± 0.02 | 0.34 |
| Basis Sharing | **32** | 286 | | 0.29 ± 0.01 | 0.52 ± 0.01 | **0.43** ± 0.01 | 0.63 ± 0.01 | **0.60** ± 0.01 | **0.31** ± 0.02 | 0.46 |
| GFWSVD (Ours) | 35 | **58** | | **0.33** ± 0.01 | **0.61** ± 0.01 | 0.42 ± 0.01 | **0.71** ± 0.01 | 0.58 ± 0.01 | 0.23 ± 0.02 | **0.48** |
| FWSVD | 11072 | 15376 | | 0.21 ± 0.01 | 0.27 ± 0.01 | 0.26 ± 0.01 | 0.54 ± 0.01 | 0.48 ± 0.01 | 0.16 ± 0.02 | 0.32 |
| ASVD | 2992 | 13193 | 40% | 0.23 ± 0.01 | 0.27 ± 0.01 | 0.26 ± 0.01 | 0.55 ± 0.01 | 0.49 ± 0.01 | 0.15 ± 0.02 | 0.33 |
| Basis Sharing | 78 | 1083 | | 0.24 ± 0.01 | 0.39 ± 0.01 | **0.33** ± 0.01 | 0.56 ± 0.01 | **0.56** ± 0.01 | **0.28** ± 0.02 | **0.39** |
| GFWSVD (Ours) | **69** | **101** | | **0.25** ± 0.01 | **0.41** ± 0.01 | 0.32 ± 0.01 | **0.61** ± 0.01 | 0.55 ± 0.01 | 0.22 ± 0.02 | **0.39** |
| FWSVD | 18992 | 23088 | | 0.20 ± 0.01 | 0.27 ± 0.01 | 0.26 ± 0.01 | 0.50 ± 0.01 | 0.51 ± 0.01 | 0.15 ± 0.02 | 0.31 |
| ASVD | 4039 | 46189 | 50% | 0.22 ± 0.01 | 0.26 ± 0.01 | 0.26 ± 0.01 | 0.50 ± 0.01 | 0.48 ± 0.01 | 0.13 ± 0.02 | 0.31 |
| Basis Sharing | 203 | 3506 | | 0.23 ± 0.01 | 0.30 ± 0.01 | **0.29** ± 0.01 | 0.52 ± 0.01 | 0.53 ± 0.01 | **0.26** ± 0.02 | 0.35 |
| GFWSVD (Ours) | **176** | **501** | | **0.24** ± 0.01 | **0.31** ± 0.01 | 0.28 ± 0.01 | **0.55** ± 0.01 | **0.54** ± 0.01 | 0.22 ± 0.02 | **0.36** |

*Table 4.* Throughput (tokens/s) achieved by the uncompressed Llama 2 7B Chat and its FWSVD-compressed versions.

| C. Ratio | Tokens/s | Relative Speedup |
|---|---|---|
| 0% | 1186 | 1.00× |
| 10% | 1269 | 1.07× |
| 20% | 1323 | 1.12× |
| 40% | 1510 | 1.27× |
| 50% | 1600 | 1.34× |

surpasses Basis Sharing by nearly 10%, whereas on Open-BookQA the opposite pattern emerges. This suggests that different types of structural dependencies within the model – parameter-level versus inter-layer relationships – benefit different categories of tasks.

More fine-grained compression results at 5–20% compression and MMLU evaluation are provided in Appendix I and Figure 7. There, we show that as the compression ratio decreases, the relative importance of diagonal Fisher information grows, and GFWSVD increasingly outperforms both FWSVD and ASVD. To demonstrate how model compression accelerates inference, we report throughput speedups in Table 4 (averaged over 100 runs, batch size = 1, sequence length = 1024 tokens, GPU: A100 80GB). The corresponding FLOP counts are detailed in Appendix F. Code for experiments is available at `https://github.com/sayankotor/FisherKronecker`.

### 6.4. Integration with Optimization Pipelines

GFWSVD can initialize optimization pipelines with subsequent fine-tuning of decomposition parameters. It can be seamlessly integrated into any such pipeline as a drop-in re-placement for standard SVD, since its factors are computed once before training. We integrated GFWSVD into the Dobi-SVD pipeline to fine-tune Llama 2 7B for 20 epochs. Evaluated on ARC, PIQA, and HellaSwag (Table 6), Dobi-GFWSVD consistently outperformed the original baseline. At 20% compression, it maintains competitive performance with only a 3% perplexity increase; at 40%, it remains robust, showing only moderate degradation.

## 7. Conclusion

In summary, we addressed three interconnected objectives: (1) introduced Matrix-free Fisher Factorization, a GPU-tractable Kronecker approximation of the Fisher matrix; (2) proved that under MVN and Kronecker-structured Fisher assumptions GFWSVD is the unique optimal Fisher-weighted low-rank decomposition; (3) demonstrated that GFWSVD yields state-of-the-art compression and robust initialization across various neural network architectures.

By avoiding explicit construction of the full FIM matrix, MFF enables efficient capture of both row-wise and column-wise parameter correlations, providing a practical pathway to leveraging second-order information in large neural networks. We established a theoretical connection between Fisher geometry and optimal low-rank approximation. Under the Matrix-Variate Normal assumptions, we proved that the resulting GFWSVD approach yields the unique optimal post-training low-rank decomposition that minimizes the expected second-order loss increase in neural network compression tasks. This result generalizes prior diagonal Fisher-based methods (FWSVD) and provides a theoretically grounded foundation for Fisher-weighted parameter

*Table 5.* Performance of the Llama 2 7B Chat compressed by various methods under compression ratios from 20% to 50% on WikiText 2 (WT2) and six common sense reasoning datasets. Lower is better for perplexity (↓), higher is better for accuracy (↑).

| METHOD | WT2↓ | C. Ratio | ARC-C↑ | ARC-E↑ | HellaSwag↑ | PIQA↑ | WinoGrande↑ | OpenBook↑ | AVG↑ |
|---|---|---|---|---|---|---|---|---|---|
| Full model | **6.94** | 0% | **0.44** ± 0.01 | **0.73** ± 0.01 | **0.58** ± 0.01 | **0.76** ± 0.01 | **0.67** ± 0.01 | **0.33** ± 0.02 | **0.59** |
| FWSVD | 66.18 | | 0.24 ± 0.01 | 0.48 ± 0.01 | 0.38 ± 0.01 | 0.64 ± 0.01 | 0.58 ± 0.01 | 0.18 ± 0.02 | 0.42 |
| ASVD | 18.33 | | 0.27 ± 0.01 | 0.51 ± 0.01 | 0.39 ± 0.01 | 0.68 ± 0.01 | 0.61 ± 0.01 | 0.22 ± 0.02 | 0.45 |
| SVD-LLM | 12.10 | 20% | 0.29 ± 0.01 | **0.66** ± 0.01 | 0.40 ± 0.01 | 0.66 ± 0.01 | **0.61** ± 0.01 | 0.23 ± 0.02 | 0.48 |
| Basis Sharing | **11.1** | | 0.31 ± 0.01 | 0.65 ± 0.01 | 0.42 ± 0.01 | 0.68 ± 0.01 | **0.61** ± 0.01 | **0.27** ± 0.02 | 0.49 |
| GFWSVD (Ours) | **11.1** | | **0.33** ± 0.01 | 0.62 ± 0.01 | **0.47** ± 0.01 | **0.74** ± 0.01 | **0.61** ± 0.01 | 0.25 ± 0.02 | **0.50** |
| FWSVD | 2572 | | 0.24 ± 0.01 | 0.32 ± 0.01 | 0.27 ± 0.01 | 0.58 ± 0.01 | 0.51 ± 0.01 | 0.17 ± 0.02 | 0.35 |
| ASVD | 97.68 | | 0.21 ± 0.01 | 0.31 ± 0.01 | 0.29 ± 0.01 | 0.63 ± 0.01 | 0.54 ± 0.01 | 0.15 ± 0.02 | 0.36 |
| SVD-LLM | 18.29 | 30% | 0.25 ± 0.01 | 0.52 ± 0.01 | 0.34 ± 0.01 | 0.62 ± 0.01 | 0.55 ± 0.01 | 0.22 ± 0.02 | 0.42 |
| Basis Sharing | 15.40 | | 0.27 ± 0.01 | **0.58** ± 0.01 | 0.38 ± 0.01 | **0.63** ± 0.01 | 0.58 ± 0.01 | **0.26** ± 0.02 | **0.45** |
| GFWSVD (Ours) | **13.92** | | **0.28** ± 0.01 | 0.56 ± 0.01 | **0.40** ± 0.01 | **0.63** ± 0.01 | 0.58 ± 0.01 | 0.20 ± 0.02 | 0.44 |
| FWSVD | 9286 | | 0.23 ± 0.01 | 0.26 ± 0.01 | 0.25 ± 0.01 | 0.48 ± 0.01 | 0.45 ± 0.01 | 0.16 ± 0.02 | 0.31 |
| ASVD | 2992 | | 0.22 ± 0.01 | 0.26 ± 0.01 | 0.26 ± 0.01 | 0.49 ± 0.01 | 0.49 ± 0.01 | 0.16 ± 0.02 | 0.31 |
| SVD-LLM | 25.16 | 40% | 0.26 ± 0.01 | 0.45 ± 0.01 | 0.30 ± 0.01 | 0.55 ± 0.01 | 0.54 ± 0.01 | 0.19 ± 0.02 | 0.38 |
| Basis Sharing | 17.26 | | 0.21 ± 0.01 | 0.46 ± 0.01 | 0.32 ± 0.01 | 0.58 ± 0.01 | 0.55 ± 0.01 | **0.19** ± 0.02 | 0.39 |
| GFWSVD (Ours) | **16.70** | | **0.27** ± 0.01 | **0.48** ± 0.01 | **0.33** ± 0.01 | **0.64** ± 0.01 | **0.57** ± 0.01 | 0.17 ± 0.02 | **0.41** |
| FWSVD | 36578 | | 0.22 ± 0.01 | 0.25 ± 0.01 | 0.25 ± 0.01 | 0.52 ± 0.01 | 0.50 ± 0.01 | **0.17** ± 0.02 | 0.32 |
| ASVD | 16896 | | 0.21 ± 0.01 | 0.25 ± 0.01 | 0.26 ± 0.01 | 0.53 ± 0.01 | 0.49 ± 0.01 | 0.16 ± 0.02 | 0.32 |
| SVD-LLM | 56.72 | 50% | 0.21 ± 0.01 | 0.33 ± 0.01 | 0.26 ± 0.01 | 0.54 ± 0.01 | 0.50 ± 0.01 | 0.12 ± 0.02 | 0.33 |
| Basis Sharing | **35.12** | | 0.20 ± 0.01 | **0.36** ± 0.01 | **0.30** ± 0.01 | 0.55 ± 0.01 | 0.50 ± 0.01 | 0.15 ± 0.02 | **0.34** |
| GFWSVD (Ours) | 37.80 | | **0.22** ± 0.01 | 0.28 ± 0.01 | 0.26 ± 0.01 | **0.55** ± 0.01 | **0.51** ± 0.01 | 0.15 ± 0.02 | 0.33 |

*Table 6.* Performance of the Llama-2 7B Chat model compressed with the Dobi-SVD and Dobi-GFWSVD at 20% and 40% ratio.

| METHOD | C. Ratio | WT2↓ | PTB↓ | ARC-E↑ | ARC-C↑ | PIQA↑ | HSwag↑ |
|---|---|---|---|---|---|---|---|
| Full model | 0% | **6.94** | **25.75** | **0.73** | **0.44** | **0.78** | **0.57** |
| Dobi-SVD | 20% | 7.75 | 26.11 | 0.71 | 0.40 | 0.76 | 0.55 |
| Dobi-GFWSVD | | **7.56** | **25.95** | **0.72** | **0.42** | **0.77** | 0.55 |
| Dobi-SVD | 40% | 10.56 | 41.70 | 0.58 | 0.32 | 0.69 | 0.32 |
| Dobi-GFWSVD | | **10.29** | **38.56** | **0.60** | **0.33** | 0.69 | 0.32 |

sensitivity and curvature-aware compression.

We conducted comprehensive experiments on synthetic benchmarks, encoder-based models, and large decoder-only language models. Our results demonstrate that GFWSVD consistently outperforms diagonal and activation-based baselines across a wide range of compression regimes. Moreover, when used as an initialization for optimization pipelines, GFWSVD improves fine-tuning robustness under aggressive compression compared to standard initialization. While our empirical evaluation focuses on an instance of post-training compression tasks, the underlying formulation directly targets parameter sensitivity via the Fisher-weighted quadratic loss model, and the same primitives apply to other sensitivity-driven applications. Together, these results position MFF and GFWSVD as foundational primitives for scalable, second-order-aware neural network approximation and parameter sensitivity.

## Impact Statement

This work develops scalable methods for analyzing parameter sensitivity in neural networks through a curvature-aware, Fisher-guided framework. Our research does not involve human subjects or sensitive data. All experiments use publicly available models and standard benchmarks. By enabling efficient approximation of second-order information, our method may reduce the computational and energy costs of large model deployment. As with any efficiency technique, we encourage responsible use and appropriate safeguards in downstream applications.

## Acknowledgements

This work was supported by the Ministry of Economic Development of the Russian Federation in accordance with the subsidy agreement (agreement identifier 000000C313925P4H0002; grant No 139-15-2025-012).

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

# A. Appendix: Limitations

**Local optimality vs. stochastic training.** Our analysis assumes a fixed optimum $W^\star$ and optimizes the second-order loss increase around this point, i.e., the Fisher-weighted quadratic model in Eq. (2). In practice, large neural networks are trained with stochastic optimizers, implicit regularization, and ongoing fine-tuning, so the realized weights may deviate from the ideal MLE solution. Consequently, GFWSVD is guaranteed optimal only for this local quadratic approximation, not for the full non-stationary training dynamics.

**Matrix-variate normal modeling of weights.** The optimality result in Theorem 3.1 is derived under the assumption that the layer weights follow a Matrix-Variate Normal distribution $\mathbf{W} \sim \mathcal{MN}(\mathbf{W}^\star, \mathbf{B}^{-1}, \mathbf{A}^{-1})$, whose covariance is aligned with the Kronecker-structured Fisher. This MVN assumption is analytically convenient and provides a sufficient condition for our closed-form solution, but it is unlikely to hold exactly in modern deep networks, where weight distributions may exhibit heavy tails or multimodality. Other perturbation models could in principle yield different "optimal" decompositions for the true perturbation distribution; nonetheless, our experiments show that the MVN-based formulation consistently outperforms alternative intra-layer methods and often remains competitive even with more elaborate cross-layer schemes.

**Cross-layer coordination** . We observed that compression effectiveness varies significantly across layers, making preliminary layer selection necessary to achieve favorable trade-offs. A key limitation of our current approach is the lack of coordination across layers during compression. For effective multi – layer compressionespecially in large-scale models like LLMs – it is important to account for cross-layer dependencies. Future work could focus on modeling these interactions to enable joint compression strategies.

# B. Appendix: Proof of the Theorem 3.1

*Proof.* Under the assumption that the loss function originates from MLE, the Hessian coincides with Fisher Information at the optimal point, ensuring structured sensitivity encoding. Hence, one can replace Eq. 2 with a surrogate problem

$$\min_{\mathcal{C}} \ (\theta^\star - \mathcal{C}(\theta^\star))^\top \mathcal{I}_F (\theta^\star - \mathcal{C}(\theta^\star)) \tag{13}$$

for $\text{vec}(\mathbf{W}^\star) = \theta^\star$ and $\text{vec}(\mathbf{W}) = \mathcal{C}(\theta^\star)$.

Substituting $\mathcal{I}_F$ with $\mathbf{A} \otimes \mathbf{B}$ and applying Cholesky decomposition to factors $\mathbf{A}$ and $\mathbf{B}$ yields:

$$\text{vec}(\mathbf{W}^\star - \mathbf{W})^\top (\mathbf{L_A L_A}^\top \otimes \mathbf{L_B L_B}^\top) \text{vec}(\mathbf{W}^\star - \mathbf{W})$$
$$= \text{vec}(\mathbf{W}^\star - \mathbf{W})^\top (\mathbf{L_A} \otimes \mathbf{L_B})(\mathbf{L_A}^\top \otimes \mathbf{L_B}^\top) \text{vec}(\mathbf{W}^\star - \mathbf{W})$$
$$= \text{vec}(\mathbf{L_B}^\top (\mathbf{W}^\star - \mathbf{W})\mathbf{L_A})^\top \text{vec}(\mathbf{L_B}^\top (\mathbf{W}^\star - \mathbf{W})\mathbf{L_A})$$
$$= \left\| \mathbf{L_B}^\top (\mathbf{W}^\star - \mathbf{W}) \mathbf{L_A} \right\|_F^2 \tag{14}$$

In Section 3.2, we established that under the assumption $\mathbf{W} \sim \mathcal{MN}(\mathbf{W}^\star, \mathbf{B}^{-1}, \mathbf{A}^{-1})$ the optimal solution to this problem can be obtained via the standard SVD of the auxiliary matrix $\widetilde{\mathbf{W}}$. The final solution is found in two steps: 1) finding an optimal rank-$r$ solution to the auxiliary problem $\widetilde{\mathbf{W}}_r = \text{SVD}_r(\mathbf{L_B}^\top \mathbf{W}^\star \mathbf{L_A})$, and 2) recovering the optimal solution to the original problem through the inverse transformation $\widehat{\mathbf{W}}_r = \mathbf{L_B}^{-\top} \widetilde{\mathbf{W}}_r \mathbf{L_A}^{-1}$, which yields the best rank-$r$ minimizer for Eq. 14. Consequently, the decomposition $\widehat{\mathbf{W}}_r$ presents an optimal compression $\mathcal{C}$ for Eq. 13, which in turn yields the minimal error increase in Eq. 1 for the given task defined by Eq. 2. ∎

# C. Appendix: Additional explanations for Kronecker decomposition adaptation

Let's show that the permuted $\mathcal{I}_F$ in the Kronecker decomposition algorithm can be expressed as the Kronecker product of the corresponding gradient matrices.

We start with the empirical Fisher information matrix defined as $\mathcal{I}_F = \frac{1}{|D|} \sum_{i=1}^{|D|} g_i g_i^\top$ and its reordered version:

$$\tilde{\mathcal{I}}_F = \mathcal{R}\mathcal{I}_F \tag{15}$$

Using the identity

$$\mathrm{vec}(\mathbf{g}_i \mathbf{g}_i^\top) = \mathbf{g}_i \otimes \mathbf{g}_i,$$

we obtain:

$$\mathrm{vec}(\mathcal{I}_F) = \frac{1}{|D|} \sum_{i=1}^{|D|} \mathrm{vec}(\mathbf{g}_i \mathbf{g}_i^\top) = \frac{1}{|D|} \sum_{i=1}^{|D|} (\mathbf{g}_i \otimes \mathbf{g}_i). \tag{16}$$

Let $\mathcal{P} \in \mathbb{R}^{(ab)^2 \times (ab)^2}$ be the unique permutation matrix such that for any matrices $\mathbf{A}, \mathbf{B} \in \mathbb{R}^{a \times b}$:

$$\mathcal{P} \cdot \mathrm{vec}(\mathbf{A} \otimes \mathbf{B}) = (\mathrm{vec}(\mathbf{A}) \otimes \mathrm{vec}(\mathbf{B})). \tag{17}$$

In our case $\mathcal{P}$ can be defined through the commutation matrix $\mathbf{K}_{mn}$ and identity matrices $\mathbf{I}_n$ and $\mathbf{I}_m$:

$$\mathbf{P} := \mathbf{I}_n \otimes \mathbf{K}_{mn} \otimes \mathbf{I}_m, \qquad \mathbf{K}_{mn}^\top = \mathbf{K}_{nm} \tag{18}$$

Using this , we can write:

$$\mathcal{P} \cdot \mathrm{vec}(\mathbf{G}_i \otimes \mathbf{G}_i) = \mathrm{vec}(\mathbf{G}_i) \otimes \mathrm{vec}(\mathbf{G}_i). \tag{19}$$

Therefore, the vectorized Fisher information becomes:

$$\mathrm{vec}(\mathcal{I}_F) = \frac{1}{|D|} \sum_{i=1}^{|D|} \mathcal{P} \cdot \mathrm{vec}(\mathbf{G}_i \otimes \mathbf{G}_i) = \mathcal{P} \cdot \mathrm{vec}\left( \frac{1}{|D|} \sum_{i=1}^{|D|} (\mathbf{G}_i \otimes \mathbf{G}_i) \right) = \mathcal{P} \, \mathrm{vec}(\tilde{\mathcal{I}}_F). \tag{20}$$

So, $\tilde{\mathcal{I}}_F$ can be defined as $\frac{1}{|D|} \sum_{i=1}^{|D|} (\mathbf{G}_i \otimes \mathbf{G}_i)$. This fact is used in the accelerated adaptation of the Kronecker Factorization algorithm.

Now, suppose a $\mathcal{I}_F$ and $\tilde{\mathcal{I}}_F$ are connected with $\mathcal{R} \in \mathbb{R}^{n \times n}$ (see Eq. 15):

$$\mathrm{vec}(\widetilde{\mathcal{I}}_F) = (I \otimes \mathcal{R}) \cdot \mathrm{vec}(\mathcal{I}_F), \mathcal{P} = I \otimes \mathcal{R} \tag{21}$$

## D. Appendix: Right vector-matrix multiplication

We can define right vector-matrix multiplication as follows:

$$\mathcal{I}_F^\top z = (\sum_{i=1}^{|D|} \mathbf{G}_i \otimes \mathbf{G}_i)^\top z \tag{22}$$

Using property of the Kronecker product $(\mathbf{K} \otimes \mathbf{L}) \, \mathrm{vec}(\mathbf{C}) = \mathrm{vec}(\mathbf{K}^\top \mathbf{C} \mathbf{L})$:

$$\mathcal{I}_F^\top z = \sum_{i=1}^{|D|} \mathrm{vec}(\mathbf{G}_i \mathbf{Z} \mathbf{G}_i^\top), \text{ where } z = \mathrm{vec}(\mathbf{Z}), \mathbf{Z} \in \mathbb{R}^{m \times m} \tag{23}$$

## E. Appendix: Special case of diagonal Fisher Information Matrix

In this section, we show that FWSVD (Hsu et al., 2022) is a special case of our generalized approach.

Hsu et al. (2022) propose to minimize the following objective:

$$\min_{\mathbf{W_1}, \mathbf{W_2}} \|\mathbf{D}\mathbf{W}^\star - \mathbf{D}\mathbf{W_2}\mathbf{W_1}\|_F^2 \tag{24}$$

where $\mathbf{D}$ is the diagonal matrix $\sqrt{\mathrm{diag}\left(\mathbb{E}[\mathbf{G}\mathbf{G}^\top]\right)}$. Specifically, $\mathbf{D}_{i,i} = \sqrt{\sum_{j=1}^{m}\mathbb{E}(\mathbf{G}_{i,j})^2}$.

Similarly to 9, we approximate the Fisher Information with a Kronecker product of identity matrix $\mathbf{I}_m$ and some diagonal matrix $\tilde{\mathbf{D}}$. As described further in Section 4 and Appendix E, under the permutation $\mathcal{R}$, the problem

$$\min_{\mathbf{D}} \left\| \mathbf{I}_F - \mathbf{I}_m \otimes \tilde{\mathbf{D}} \right\|_{\mathrm{F}} \tag{25}$$

reduces to minimization of the expression

$$\min_{\mathbf{d}} \left\| \mathbb{E}[\mathbf{G} \otimes \mathbf{G}] - (\mathbf{I}_n \odot \mathbf{I}_n)d \cdot \mathrm{vec}(\mathbf{I}_m)^\top \right\|_{\mathrm{F}} \tag{26}$$

where $\odot$ is a Khatri-Rao product (column-wise Kronecker product) and $\cdot$ is a vector outer product; $d$ is a vector diagonal of $\tilde{\mathbf{D}}$; $\mathbb{E}[\mathbf{G} \otimes \mathbf{G}]$ is a permuted Fisher Information matrix $\tilde{\mathbf{I}}_{\mathbf{F}}$, defined in Eq 10.

For simplicity, we will use a shorter notation. Let $\mathbf{E} = \mathbb{E}[\mathbf{G} \otimes \mathbf{G}]$, $\mathbf{Z} = \mathbf{I}_n \odot \mathbf{I}_n$, $v = \mathrm{vec}(\mathbf{I}_m)$. Then, the problem 26 is equivalent to

$$\min_{\mathbf{d}} \left\| \mathbf{Z}d \cdot v^\top - \mathbf{E} \right\|_{\mathrm{F}} \tag{27}$$

Applying first-order optimality conditions yields:

$$\langle \mathbf{Z}\delta d \cdot v^\top, \mathbf{Z}d \cdot v^\top - \mathbf{E} \rangle = 0$$
$$\langle \delta d \cdot v^\top, \mathbf{Z}^\top \mathbf{Z}d \cdot v^\top - \mathbf{Z}^\top \mathbf{E} \rangle = 0$$
$$\langle \delta d, \mathbf{Z}^\top \mathbf{Z}d \cdot v^\top v - \mathbf{Z}^\top \mathbf{E}v \rangle = 0$$

Since $\mathbf{Z}^\top \mathbf{Z} = \mathbf{I}_n$, $v^\top v = \|v\|_2^2 = \|\mathrm{vec}(\mathbf{I}_m)\|_2^2 = m$ , we have:

$$d = \frac{1}{m}(\mathbf{I}_n \odot \mathbf{I}_n)^\top \mathbb{E}[\mathbf{G} \otimes \mathbf{G}]\mathrm{vec}(\mathbf{I}_m) = \frac{1}{m}(\mathbf{I}_n \odot \mathbf{I}_n)^\top \mathrm{vec}(\mathbb{E}[GG^\top]) = \frac{1}{m}\mathrm{diag}(\mathbb{E}[GG^\top]) \tag{28}$$

Thus, diagonal matrix $\tilde{\mathbf{D}}$ from Kronecker product approximation problem 25 equals square of matrix $\mathbf{D}$ from the FWSVD formulation 24 up to the constant $\frac{1}{m}$.

We apply Theorem 3.1 to find factors $\mathbf{W}_2$, $\mathbf{W}_1$ for the obtained approximation $\mathbf{I}_F = \mathbf{I}_m \otimes \tilde{\mathbf{D}}$:

$$\mathbf{W}_2 = \sqrt{\tilde{\mathbf{D}}}^{-1}\hat{\mathbf{U}}_r\sqrt{\hat{\mathbf{S}}_r} = \mathbf{D}^{-1}\hat{\mathbf{U}}_r\sqrt{\hat{\mathbf{S}}_r}, \mathbf{W}_1 = \sqrt{\hat{\mathbf{S}}_r}\hat{\mathbf{V}}_r^\top \tag{29}$$

where $\hat{\mathbf{U}}_r\hat{\mathbf{S}}_r\hat{\mathbf{V}}_r^\top$ is r-rank SVD of $\sqrt{\tilde{\mathbf{D}}}\mathbf{W}^\star = \mathbf{D}\mathbf{W}^\star$. This is the same solution that minimizes the problem 24 from FWSVD paper (Hsu et al., 2022). Consequently, FWSVD approach is a special case of diagonal Kronecker product approximation of Fisher Information.

## F. Appendix: FLOPs for compressed models

Methods like FWSVD, ASVD, SVD-LLM and our GFWSVD compress a dense weight matrix $\mathbf{W} \in \mathbb{R}^{n \times m}$ by decomposing it into two low-rank factors $\mathbf{W}_1 \in \mathbb{R}^{n \times r}$ and $\mathbf{W}_2 \in \mathbb{R}^{r \times m}$, where $r \ll \min(n, m)$. This decomposition reduces the computational complexity of a forward pass from $\mathcal{O}(nm)$ to $\mathcal{O}(r(n + m))$. In this section we describe the additional quantification of this theoretical reduction for Llama 2 7B in Table 7.

To assess the practical realization of these gains, we measured end-to-end inference latency on a single NVIDIA A100 80GB GPU. We report the average time per token over 100 runs with a sequence length of 1024. As shown in Table 8, GFWSVD achieves consistent speedups, reaching a $1.34\times$ improvement at 50% reduction.

## G. Appendix: Extended GLUE results

We report extended compression results on tasks of GLUE benchmark in Table 9.

*Table 7.* Comparison of theoretical FLOPs for Llama 2 7B Chat under different compression rates. All values are in trillions (T) of FLOPs.

| Model | Compression Ratio | Full Model FLOPs | Compressed FLOPs |
|---|---|---|---|
| | 10% | 53.05T | 42.43T |
| | 15% | 53.05T | 39.24T |
| Llama 2 7B | 20% | 53.05T | 37.18T |
| | 40% | 53.05T | 31.83T |
| | 50% | 53.05T | 29.37T |

*Table 8.* Inference latency (in milliseconds) per token for compressed Llama 2 7B Chat model. Lower is better. Reported values represent forward pass time averaged over 100 runs.

| Method | Compression Ratio | | | | | |
|---|---|---|---|---|---|---|
| | 0% | 10% | 15% | 20% | 40% | 50% |
| | Throughput↓ (`batch_size=64`) | | | | | |
| GFWSVD (Ours) | 4.7 | 4.5 | 4.2 | 4.0 | 3.3 | 2.95 |
| SVD-LLM | 4.7 | 4.4 | 4.2 | 3.9 | – | – |
| | Throughput↓ (`batch_size=16`) | | | | | |
| GFWSVD (Ours) | 3.2 | 3.0 | 2.8 | 2.6 | 2.3 | 2.15 |
| SVD-LLM | 3.2 | 2.9 | 2.8 | 2.6 | – | |

## H. Appendix: Impact of Diagonal and Non-diagonal Elements of Factors

To assess the significance of off-diagonal correlations in Transformer architectures, we performed the following ablation study. We compressed the Llama-2-Chat model using the GFWSVD method by retaining either: (1) only the off-diagonal elements (**Non-diag**), effectively nulling the diagonal, or (2) only the diagonal elements (**Diag**), and measured perplexity relative to GFWSVD and FWSVD. The results are in Table 10: the **Diag** variant performs better than FWSVD but worse than GFWSVD. This is expected, since FWSVD captures importance only along rows (only the left factor matrix has a non-identity diagonal, see Fig. 1), whereas Non-diag GFWSVD captures both row and column importance. The contribution of off-diagonal elements provides a noticeable improvement compared to FWSVD.

## I. Appendix: Extended Decoder Evaluation on MMLU

Table 11 and Figure 7 show that for Llama 2 7B GFWSVD consistently outperforms both simple and strong baselines across all compression rates. In particular, at the most aggressive compression setups (15–20% of the original parameters), our method matches or exceeds the accuracy of activation-based methods and shows substantially lower perplexities on both WikiText-2 and PTB.

We also compressed the Llama 3.1 8B model using our GFWSVD and compared it to the activation-aware SVD (ASVD) method. Due to its extensive training on 15 trillion tokens, Llama 3.1 has exceptionally high information density and low parameter redundancy, therefore, it is a significantly more challenging target for compression than Llama 2. In Table 12 we show that Llama 3.1 has a stronger degradation in quality upon compression than Llama 2. Nevertheless, our method GFWSVD demonstrated better results across all compression ratios.

## J. Appendix: LLM Usage Statement

We used large language models (LLMs) only as a general-purpose writing assistant for grammar checking and text polishing. The research ideas, implementation, analysis, and conclusions are entirely our own.

*Table 9.* Performance of BERT model compressed by various methods under compression rates from 60% to 99% on GLUE benchmark. Higher is better for all tasks (↑).

| METHOD / DATASET | MRPC↑ | STSB↑ | QQP↑ | MNLI↑ | QNLI↑ | RTE↑ | COLA↓ | SST2↑ |
|---|---|---|---|---|---|---|---|---|
| Full model | **0.77** | **0.87** | **0.90** | **0.83** | **0.90** | **0.56** | **0.41** | **0.91** |
| Compression Ratio 1% ($r = 600$) | | | | | | | | |
| SVD | 0.67 | 0.84 | 0.90 | 0.67 | 0.90 | 0.56 | 0.58 | 0.91 |
| ASVD (Yuan et al., 2023) | 0.72 | 0.73 | 0.89 | 0.83 | 0.90 | 0.56 | 0.41 | 0.91 |
| FWSVD (Hsu et al., 2022) | 0.72 | 0.87 | 0.90 | 0.72 | 0.90 | 0.55 | 0.36 | 0.91 |
| GFWSVD (Ours) | **0.73** | **0.87** | **0.90** | **0.73** | **0.90** | **0.56** | **0.55** | **0.92** |
| Compression Ratio 8% ($r = 500$) | | | | | | | | |
| SVD | 0.53 | 0.82 | 0.89 | 0.53 | 0.90 | 0.54 | 0.53 | 0.89 |
| ASVD (Yuan et al., 2023) | 0.71 | 0.56 | 0.86 | 0.81 | 0.89 | 0.53 | 0.44 | 0.88 |
| FWSVD (Hsu et al., 2022) | 0.71 | 0.87 | 0.90 | 0.71 | 0.89 | 0.56 | 0.34 | 0.91 |
| GFWSVD (Ours) | **0.73** | **0.87** | **0.90** | **0.73** | **0.90** | **0.56** | **0.49** | **0.92** |
| Compression Ratio 23% ($r = 250$) | | | | | | | | |
| SVD | 0.49 | 0.68 | 0.81 | 0.49 | 0.85 | 0.50 | 0.17 | 0.57 |
| ASVD (Yuan et al., 2023) | 0.69 | 0.08 | 0.76 | 0.50 | 0.58 | 0.47 | 0.11 | 0.75 |
| FWSVD (Hsu et al., 2022) | 0.69 | 0.86 | 0.89 | 0.69 | 0.89 | 0.61 | 0.23 | 0.80 |
| GFWSVD (Ours) | **0.71** | **0.86** | **0.89** | **0.71** | **0.89** | **0.61** | **0.38** | **0.88** |
| Compression Ratio 33% ($r = 100$) | | | | | | | | |
| SVD | 0.32 | 0.08 | 0.64 | 0.32 | 0.80 | 0.51 | 0.01 | 0.49 |
| ASVD (Yuan et al., 2023) | 0.58 | 0.07 | 0.74 | 0.39 | 0.50 | 0.47 | 0.05 | 0.82 |
| FWSVD (Hsu et al., 2022) | 0.69 | 0.58 | 0.87 | 0.71 | 0.86 | 0.55 | 0.21 | 0.72 |
| GFWSVD (Ours) | **0.71** | **0.70** | **0.87** | **0.71** | **0.86** | **0.55** | **0.21** | **0.72** |
| Compression Ratio 36% ($r = 50$) | | | | | | | | |
| SVD | 0.32 | 0.19 | 0.57 | 0.32 | 0.78 | 0.48 | 0.02 | 0.49 |
| ASVD (Yuan et al., 2023) | 0.68 | 0.03 | 0.73 | 0.49 | 0.76 | 0.51 | 0.03 | 0.80 |
| FWSVD (Hsu et al., 2022) | 0.69 | 0.65 | 0.84 | 0.69 | 0.72 | 0.46 | 0.03 | 0.77 |
| GFWSVD (Ours) | **0.69** | **0.65** | **0.84** | **0.69** | **0.72** | **0.46** | **0.05** | **0.77** |
| Compression Ratio 39% ($r = 10$) | | | | | | | | |
| SVD | 0.32 | 0.32 | 0.67 | 0.32 | 0.61 | 0.51 | 0.00 | 0.49 |
| ASVD (Yuan et al., 2023) | 0.61 | 0.14 | 0.64 | 0.40 | 0.57 | 0.49 | -0.04 | 0.76 |
| FWSVD (Hsu et al., 2022) | 0.37 | 0.32 | 0.79 | 0.37 | 0.57 | 0.49 | 0.00 | 0.49 |
| GFWSVD (Ours) | **0.53** | **0.60** | **0.79** | **0.53** | **0.62** | **0.47** | **0.05** | **0.65** |
| Compression Ratio 40% ($r = 1$) | | | | | | | | |
| SVD | 0.32 | 0.04 | 0.69 | 0.31 | 0.55 | 0.53 | 0.00 | 0.49 |
| ASVD (Yuan et al., 2023) | 0.62 | 0.10 | 0.64 | 0.42 | 0.50 | 0.49 | 0.03 | 0.70 |
| FWSVD (Hsu et al., 2022) | 0.32 | 0.18 | 0.72 | 0.32 | 0.51 | 0.50 | 0.00 | 0.49 |
| GFWSVD (Ours) | **0.42** | **0.70** | **0.74** | **0.42** | **0.65** | **0.52** | **0.05** | **0.49** |

*Table 10.* Perplexity (↓) at 90% and 85% compression rates for GFWSVD with full, diagonal-only and non-diagonal factors for Llama 2 7B Chat compression.

| METHOD / DATASET | WikiText 2↓ | PTB↓ | WikiText 2↓ | PTB↓ |
|---|---|---|---|---|
| Compression Ratio | 10% | 10% | 15% | 15% |
| FWSVD (Hsu et al., 2022) | 11.53 | 96.62 | 22.00 | 411.00 |
| Diag GFWSVD | 10.94 | 45.26 | 11.06 | 48.25 |
| Non-diag GFWSVD | 8.85 | 37.25 | 10.22 | 43.75 |
| Full GFWSVD (Ours) | **8.77** | **36.44** | **10.06** | **42.19** |

*Table 11.* Performance of the Llama 2 7B Chat compressed by various methods under compression ratios from 5% to 20% on WikiText-2 (WT2), PTB, and MMLU. Lower is better for perplexity (↓), higher is better for accuracy (↑). We denote compression ratio as $1 - \frac{\text{compressed model}}{\text{original model}}$.

| METHOD | WT2↓ | PTB↓ | C. Ratio | MMLU Avg↑ | Humanities↑ | Other↑ | Social Sciences↑ | STEM↑ |
|---|---|---|---|---|---|---|---|---|
| Full model | **6.94** | **25.75** | 0% | $0.46 \pm 0.003$ | $0.43 \pm 0.01$ | $0.55 \pm 0.01$ | $0.53 \pm 0.01$ | $0.36 \pm 0.01$ |
| FWSVD (Hsu et al., 2022) | 7.52 | 45.25 | | $0.40 \pm 0.003$ | $0.36 \pm 0.01$ | $0.45 \pm 0.01$ | $0.45 \pm 0.01$ | $0.35 \pm 0.01$ |
| ASVD (Yuan et al., 2023) | 7.60 | **26.29** | 5% | $\mathbf{0.41} \pm 0.004$ | $0.37 \pm 0.01$ | $\mathbf{0.48} \pm 0.01$ | $\mathbf{0.46} \pm 0.01$ | $\mathbf{0.35} \pm 0.01$ |
| SVD-LLM (Wang et al., 2025c) | 8.80 | 51.28 | | $0.34 \pm 0.004$ | $0.31 \pm 0.01$ | $0.38 \pm 0.01$ | $0.35 \pm 0.01$ | $0.31 \pm 0.01$ |
| GFWSVD (Ours) | **7.16** | 28.55 | | $0.40 \pm 0.003$ | $\mathbf{0.38} \pm 0.01$ | $0.47 \pm 0.01$ | $0.44 \pm 0.01$ | $0.33 \pm 0.01$ |
| FWSVD (Hsu et al., 2022) | 11.53 | 96.62 | | $0.37 \pm 0.004$ | $0.34 \pm 0.01$ | $0.43 \pm 0.01$ | $0.42 \pm 0.01$ | $0.33 \pm 0.01$ |
| ASVD (Yuan et al., 2023) | 8.97 | 40.12 | 10% | $0.37 \pm 0.004$ | $0.33 \pm 0.01$ | $0.42 \pm 0.01$ | $0.40 \pm 0.01$ | $0.33 \pm 0.01$ |
| SVD-LLM (Wang et al., 2025c) | 9.69 | 60.82 | | $0.32 \pm 0.004$ | $0.30 \pm 0.01$ | $0.35 \pm 0.01$ | $0.32 \pm 0.01$ | $0.30 \pm 0.01$ |
| GFWSVD (Ours) | **8.77** | **36.44** | | $\mathbf{0.38} \pm 0.002$ | $\mathbf{0.35} \pm 0.01$ | $\mathbf{0.44} \pm 0.01$ | $\mathbf{0.42} \pm 0.01$ | $\mathbf{0.33} \pm 0.01$ |
| FWSVD (Hsu et al., 2022) | 22.06 | 411.50 | | $0.31 \pm 0.009$ | $0.29 \pm 0.01$ | $0.34 \pm 0.01$ | $0.33 \pm 0.01$ | $0.30 \pm 0.01$ |
| ASVD (Yuan et al., 2023) | 10.91 | 83.49 | 15% | $0.32 \pm 0.003$ | $0.30 \pm 0.01$ | $0.33 \pm 0.01$ | $0.32 \pm 0.01$ | $0.30 \pm 0.01$ |
| SVD-LLM (Wang et al., 2025c) | 10.36 | 72.58 | | $0.30 \pm 0.004$ | $0.29 \pm 0.01$ | $0.34 \pm 0.01$ | $0.31 \pm 0.01$ | $0.30 \pm 0.01$ |
| GFWSVD (Ours) | **10.06** | **42.19** | | $\mathbf{0.36} \pm 0.004$ | $\mathbf{0.33} \pm 0.01$ | $\mathbf{0.41} \pm 0.01$ | $\mathbf{0.38} \pm 0.01$ | $\mathbf{0.32} \pm 0.01$ |
| FWSVD (Hsu et al., 2022) | 66.37 | 1523.00 | | $0.27 \pm 0.004$ | $0.25 \pm 0.01$ | $0.30 \pm 0.01$ | $0.28 \pm 0.01$ | $0.28 \pm 0.01$ |
| ASVD (Yuan et al., 2023) | 27.73 | 241.57 | 20% | $0.26 \pm 0.004$ | $0.25 \pm 0.01$ | $0.27 \pm 0.01$ | $0.24 \pm 0.01$ | $0.28 \pm 0.01$ |
| SVD-LLM (Wang et al., 2025c) | 11.23 | 98.91 | | $0.29 \pm 0.004$ | $0.27 \pm 0.01$ | $0.32 \pm 0.01$ | $0.29 \pm 0.01$ | $0.29 \pm 0.01$ |
| GFWSVD (Ours) | **11.13** | **50.50** | | $\mathbf{0.32} \pm 0.003$ | $\mathbf{0.30} \pm 0.01$ | $\mathbf{0.35} \pm 0.01$ | $\mathbf{0.34} \pm 0.01$ | $\mathbf{0.30} \pm 0.01$ |

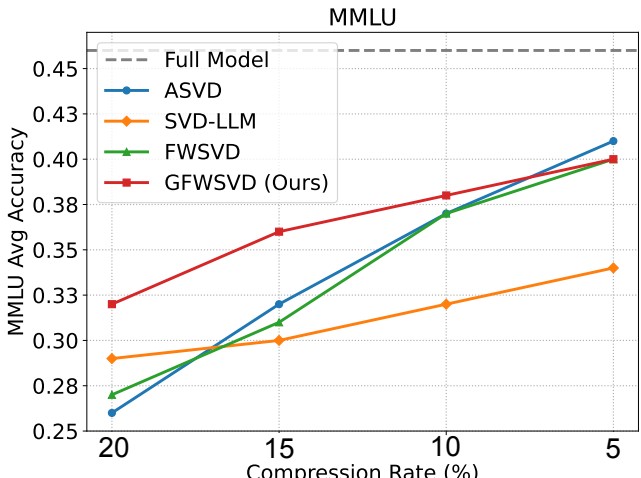

*Figure 7.* Average MMLU performance of Llama 2 model for different compression rates.

*Table 12.* Performance of Llama 3.1 8B Instruct compressed by various methods under compression ratios from 10% to 20% on WikiText-2 (WT2), PTB, and MMLU. Lower is better for perplexity ($\downarrow$), higher is better for accuracy ($\uparrow$).

| METHOD | WT2$\downarrow$ | PTB$\downarrow$ | C. Ratio | MMLU Avg$\uparrow$ | Humanities$\uparrow$ | Other$\uparrow$ | Social Sciences$\uparrow$ | STEM$\uparrow$ |
|---|---|---|---|---|---|---|---|---|
| Full model | **7.2** | **11.50** | 0% | 0.68 $\pm$ 0.006 | 0.64 $\pm$ 0.01 | 0.73 $\pm$ 0.01 | 0.78 $\pm$ 0.01 | 0.60 $\pm$ 0.01 |
| ASVD (Yuan et al., 2023) | 10.91 | **19.33** | 10% | 0.39 $\pm$ 0.004 | 0.39 $\pm$ 0.01 | 0.33 $\pm$ 0.01 | 0.35 $\pm$ 0.01 | 0.35 $\pm$ 0.01 |
| GFWSVD (Ours) | **9.38** | 19.81 | | **0.54** $\pm$ 0.002 | **0.49** $\pm$ 0.01 | **0.62**$\pm$ 0.01 | **0.63**$\pm$ 0.01 | **0.46** $\pm$ 0.01 |
| ASVD (Yuan et al., 2023) | 38.02 | 76.1 | 15% | 0.29 $\pm$ 0.004 | 0.31 $\pm$ 0.01 | 0.32 $\pm$ 0.01 | 0.31 $\pm$ 0.01 | 0.31 $\pm$ 0.01 |
| GFWSVD (Ours) | **16.75** | **23.67** | | **0.50** $\pm$ 0.001 | **0.46** $\pm$ 0.01 | **0.56** $\pm$ 0.01 | **0.57** $\pm$ 0.01 | **0.43** $\pm$ 0.01 |
| ASVD (Yuan et al., 2023) | 145 | 1672 | 20% | 0.24 $\pm$ 0.003 | 0.27 $\pm$ 0.01 | 0.28 $\pm$ 0.01 | 0.27 $\pm$ 0.01 | 0.27 $\pm$ 0.01 |
| GFWSVD (Ours) | **22.57** | **32.4** | | **0.43** $\pm$ 0.003 | **0.39** $\pm$ 0.01 | **0.49** $\pm$ 0.01 | **0.48** $\pm$ 0.01 | **0.38** $\pm$ 0.01 |

