# OpenReview forum: "Scalable Kronecker-Factored Fisher Approximation for Neural Network Parameter Sensitivity"
_ICML.cc/2026/Conference — ICML 2026 regular_

### Official Review · Reviewer_fVWg · 2026-03-10

**Soundness:** 4
**Presentation:** 3
**Significance:** 3
**Originality:** 3
**Overall Recommendation:** 5
**Confidence:** 4

**Summary:**

The authors propose a method to approximate the diagonal block of the Fisher Information Matrix (FIM) corresponding to weight parameters of a single linear layer. By rearranging the terms, they show that obtaining the optimal (in the sense of Frobenius norm eq.5) Kronecker factorization is equivalent to computing a SVD, which can be approached using Lanczos iterations by without ever materializing the (large) matrix to be approximated. This approximation is used for model compression of transformers.

**Compliance With Llm Reviewing Policy:**

Affirmed.

**Key Questions For Authors:**

## Key
1. How does your approach compare to Koroko et al's ?

## Other
2. It should be made more clearer that this is still a block-diagonal approximation, which is less confusing/relevant in your use case, but this method of obtaining optimal Kronecker factors can also disseminate in other downstream use cases of the FIM
3. How does this compare to KFAC ? Should it be added in the competitors ?
4. How much does the assumption that the weight matrix follow a MVN distribution hold ?
5. in Koroko et al. there is no such assumption, the result holds for general matrices. Can you please clarify ?
6. Do you perform minibatch Lanczos iterations, or do you have to go through the whole training dataset at each iteration?

**Limitations:**

yes

**Strengths And Weaknesses:**

## Soundness

The approach is very sound in the sense of obtaining the rank-k approximation that minimizes eq. 5, which is a very reasonable criterion for approximating the FIM. On a personal note I already implemented the idea (Lanczos iterations to obtain Kronecker factors) some while ago, and observed a reduction in Frobenius norm of the difference to the true FIM (compared to KFAC), but I was unable to obtain any significant improvement on downstream use (not model compression) of the FIM though I did not investigate it thoroughly. Moreover, the same idea was published around the same time by Koroko et al., though with more limited experiments than the current paper, so I did not push the idea further. For model compression, this looks like a promising avenue as shown experimentally in the current paper.

## Presentation

The presentation is clear, and the text is relatively easy to follow. I would appreciate to have points 1. and 2. below clarified.

## Originality and significance

The factorization approach has already been discussed in Koroko et al. in 2023, which arguably was published in a much less visible conference, and not for model compression, so I think there is still interest for publication to ICML to share the method more broadly, provided that it is properly referenced. FIMs are used in many downstream use-cases, e.g. optimization and compression in the current paper, among others, so I am overall in favor of having the paper published and reach a wider audience.

A. Koroko, A. Anciaux-Sedrakian, I. B. Gharbia, V. Garès, M. Haddou, and Q. H. Tran, “Efficient approximations of the fisher matrix in neural networks using kronecker product singular value decomposition,” ESAIM 2023, doi: 10.1051/proc/202373218.

---

> ### Author Rebuttal · Authors · 2026-03-31
>
> Thank you for the constructive and supportive review, and especially for pointing us to Koroko et al. (2023)[1]. We agree this paper is relevant and should be cited explicitly in the revision.
>
>  > How our approach compares to Koroko et al.
>
> Indeed, both works use the classical Van Loan–Pitsianis observation that the best single-Kronecker approximation of a matrix can be obtained via an SVD of a suitably permuted form[2], and both work in a layerwise / block-diagonal Fisher setting. However, the goals and contributions differ significantly:
>
> 1) Koroko et al. focus on optimization. They use Kronecker-SVD to improve natural gradient speed and quality, testing primarily on autoencoder training benchmarks.
>
> 2) Our work focuses on post-training compression. We introduce a matrix-free procedure to estimate Fisher factors without ever storing the full matrix. We then derive a proven optimal Fisher-weighted decomposition for low-rank SVD.
>
> While Koroko et al. is an important antecedent, we address a different task (compression vs. training) and provide a different theoretical contribution. Our experiments also target modern LLMs like Llama, rather than small-scale optimization. We will clarify this and the block-diagonal nature of our approach in the revised text.
>
>
> > Whether KFAC should be added as a competitor.
>
> KFAC is definitely part of the relevant background and comparison story, however, it is not a direct compression baseline in the same sense as FWSVD, ASVD, or SVD-LLM. KFAC is primarily a training-time optimizer / preconditioner, based on a layerwise block-diagonal Kronecker approximation with moving-average updates during training. In contrast, our method performs post-training compression with no additional fine-tuning.
>
>
> > On the MVN assumption, especially relative to Koroko et al.
>
> Our approach involves two steps: first, an efficient Kronecker factorization of the Hessian, and second, its application to SVD-based weight compression. The MVN assumption is employed only in the second part to prove that weighting the SVD with the obtained Kronecker factors is optimal for LLM weights (Theorem 3.1).
> Koroko et al. do not need an MVN assumption because their goal is to approximate Fisher blocks for optimization; the Kronecker factorization itself is a general matrix-approximation statement.
>
>
>
> >Do we perform mini-batch Lanczos iterations, or go through the whole training dataset at each iteration?
>
> We use a calibration dataset rather than the full training set. Specifically, we adopt the FineWeb dataset [3] due to its high quality and diversity, and collect gradients on a random subsample of 9,600 samples. Gradients are computed in a mini-batch manner (batch size = 128, 75 batches in total), and the Lanczos iterations are performed over these batches.
>
> **References**
>
> [1] Koroko, Abdoulaye, et al. "Efficient approximations of the fisher matrix in neural networks using kronecker product singular value decomposition." *ESAIM: Proceedings and Surveys 73* (2023): 218-237.
>
> [2] Van Loan, Charles F., and Nikos Pitsianis. "Approximation with Kronecker products." *Linear algebra for large scale and real-time applications. Dordrecht: Springer Netherlands*, 1993. 293-314.
>
> [3] Penedo, Guilherme, et al. "The FineWeb Datasets: Decanting the Web for the Finest Text Data at Scale." *Advances in Neural Information Processing Systems 37* (2024): 30811-30849.

---

> > ### Author Rebuttal · Reviewer_fVWg · 2026-04-01
> >
> > Regarding the last comment on mini-batch (stochastic) Lanczos, I think this should appear clearly in the experiments in a camera ready version of the paper, should it be accepted. I recommend acceptance.

---

> > > ### Author Response · Authors · 2026-04-07
> > >
> > > Thank you for your review and for the positive highlights of our paper. We will clarify in the revised version that the Lanczos iterations are performed over mini-batches.

---

### Official Review · Reviewer_Ek5B · 2026-03-11

**Soundness:** 3
**Presentation:** 3
**Significance:** 3
**Originality:** 3
**Overall Recommendation:** 4
**Confidence:** 4

**Summary:**

This paper introduces Matrix-free Fisher Factorization (MFF), a GPU-tractable method approximating FIM without forming it, preserving diagonal and off-diagonal correlations. Under matrix-variate normal assumptions it yields GFWSVD, a closed-form layer decomposition for compression. Experiments show up to 50% compression while maintaining or improving accuracy and stabilizing dense architectures.

**Compliance With Llm Reviewing Policy:**

Affirmed.

**Final Justification:**

The rebuttal addressed most of my concerns. I am happy to keep my score.

**Key Questions For Authors:**

See weaknesses.

**Limitations:**

Yes.

**Strengths And Weaknesses:**

Pros:

1Overall, the paper is well written and easy to follow. It extends prior work on post-training low-rank decomposition methods, including SVD (with identity transformation matrices) and FWSVD (where the left matrix is diagonal and the right matrix is identity). This work introduces GFWSVD, in which both transformation matrices are non-diagonal, providing a more general formulation.

2 The paper includes extensive experimental evaluations across several benchmarks. I appreciate the authors’ effort in conducting a broad empirical study.


Cons

1 After reading the paper, it is still unclear to me why generalizing FWSVD to non-diagonal transformation matrices is important and what specific limitations of FWSVD this generalization resolves. The paper would benefit from a clearer explanation of the practical motivation and why this extension is technically challenging.

2 As mentioned by the authors in the appendix, the theoretical analysis relies on strong assumptions, e.g., that the layer weights follow a matrix-variate normal (MVN) distribution and that the covariance has a Kronecker structure. However, the paper does not sufficiently justify why these assumptions are reasonable in practice. The only explanation provided is that they are “analytically convenient,” and the practical gains appear somewhat accidental.  I recommand the authors to provide  additional discussion explaining why these assumptions are meaningful or how they can guide practice.

3 The experimental gains appear relatively marginal in some cases. For example, in Table 2 (the main performance comparison table), the improvements over the second-best method are often around 0.01–0.03 on average, which makes it difficult to assess the practical significance of the method.

Minor: Fig 1is placed in the Background and Problem Formulation section (Page 3), but it is not discussed until the Relationship to Prior Works section (Page 5). Since the figure helps clarify the improvement over previous methods, it might be better to move or discuss it earlier where it first appears.

---

> ### Author Rebuttal · Authors · 2026-03-31
>
> > it is still unclear ... why generalizing FWSVD to non-diagonal transformation matrices is important and what specific limitations of FWSVD this generalization resolves
>
> The key motivation is that the diagonal Fisher assumption in FWSVD discards parameter cross-correlations that are important for capturing second-order sensitivity. In our formulation, the non-diagonal entries of the Kronecker factors ($A$ and $B$) capture correlations across rows and columns of the weight matrix, corresponding to output and input features.
>
> Thus, the main limitation of standard FWSVD is its inability to account for inter-parameter dependencies. In contrast, the full Fisher Information Matrix (FIM) provides a richer signal for low-rank approximation. The benefits of our method increase with the level of feature correlation. We demonstrate this in Section 6.1 on a synthetic correlated dataset, where our approach outperforms diagonal baselines.
>
> The prevalence of diagonal approximations in prior work is mainly due to computational constraints: the full FIM has size $nm \times nm$ for an $n \times m$ layer and is intractable. Our method addresses this by enabling a structured factorization without materializing the full matrix.
>
>
> > the theoretical analysis relies on strong assumptions, e.g., that the layer weights follow a matrix-variate normal (MVN) distribution and that the covariance has a Kronecker structure.
>
> We thank the reviewer for this comment. While we rely on strong assumptions, these are standard in machine learning and widely used in practice.
>
> **Kronecker structure**. The Kronecker factorization of the per-layer Fisher is not specific to our method; it is a standard approximation used in K-FAC[1] and related approaches, where the Fisher block is modeled as a Kronecker product of input activations and backpropagated gradient covariances [1–4]. This approximation has been empirically validated across architectures and tasks, and we adopt it for compression.
>
> **MVN as a local perturbation model.** The MVN assumption is a local model around a trained weight matrix $W^*$, not a global posterior assumption. It corresponds to the standard second-order approximation
> $\Delta L \approx \tfrac{1}{2}\mathrm{vec}(\Delta W)^\top (B \otimes A)\mathrm{vec}(\Delta W)$,
> and does not introduce additional geometric assumptions beyond local quadratic behavior. Instead, it provides a reference distribution to measure expected loss increase (consistent with [5,6]). This assumption is also standard in second-order compression and quantization methods [7,8].
>
> > The experimental gains appear relatively marginal in some cases, in Table 2.
>
> We do not claim global optimality; rather, GFWSVD is optimal within our theoretical framework, which we validate empirically. While gains may decrease when real-world structure deviates from the model, our method remains competitive with prior baselines.
> Specifically, in Table 2, we have an average improvement of ~1% over the current SOTA, which accounts for inter-layer correlations, while our method achieves these gains while compressing layer-by-layer. Under a high compression ratio, GFWSVD outperforms SOTA and maintains stability while other layer-wise methods collapse.
>
> As discussed, our method is good in correlated settings. With the current trend toward highly correlated, dense models (e.g., Llama 3.1), we believe our approach provides significant practical value for modern LLM compression.
>
> > Minor: Fig 1is placement.
>
> Thank you for this suggestion. We agree that the placement of Figure 1 should better align with its first mention and will move it to the "Relationship to Prior Works" section.
>
> **References**
>
> [1] Martens, J., & Grosse, R. Optimizing neural networks with Kronecker-factored approximate curvature. ICML, 2015.
>
> [2] Grosse, R., & Martens, J. A Kronecker-factored approximate Fisher matrix for convolution layers. ICML, 2016.
>
> [3] Tang, Z., et al. SKFAC: Training neural networks with faster Kronecker-factored approximate curvature. CVPR, 2021.
>
> [4] Dangel, F., et al. Kronecker-factored approximate curvature for physics-informed neural networks. NeurIPS, 2024.
>
> [5] Weimar, M., et al. Fisher Information Flow in Artificial Neural Networks. Phys. Rev. X, 2025.
>
> [6] Zhang, G., et al. Noisy natural gradient as variational inference. ICML, 2018.
>
> [7] Singh, S. P., & Alistarh, D. WoodFisher: Efficient second-order approximation for neural network pruning. NeurIPS, 2020.
>
> [8] Chee, J., et al. QuIP: 2-bit quantization of large language models with guarantees. NeurIPS, 2024.

---

> > ### Author Rebuttal · Reviewer_Ek5B · 2026-04-01
> >
> > “I appreciate the authors’ comprehensive response. Most of my concerns have been addressed, and I now believe the paper makes a meaningful contribution. I have increased my score to 4.

---

> > > ### Author Response · Authors · 2026-04-07
> > >
> > > We thank the reviewer for the comments and the time devoted.

---

### Official Review · Reviewer_oVfe · 2026-03-12

**Soundness:** 3
**Presentation:** 3
**Significance:** 3
**Originality:** 3
**Overall Recommendation:** 5
**Confidence:** 3

**Summary:**

The authors present a scalable approximation of the FIM, a well-known surrogate of the Hessian matrix under regular conditions. They propose MFF to approximate the empirical Fisher as a Kronecker product layer-wise, beyond diagonal approximations. Specifically, they compute the leading singular structure of a suitably permuted Fisher via efficient matrix–vector products to avoid materializing the full FIM. Under the Matrix-Variate Normal assumption, the authors derive a closed-form optimal low-rank decomposition, GFWSVD, that minimizes the expected increase in second-order loss. They validate their theoretical claims on controlled synthetic correlation settings and on Transformer models, showing that GFWSVD can serve as a strong initialization for optimization-based compression pipelines.

**Compliance With Llm Reviewing Policy:**

Affirmed.

**Final Justification:**

The rebuttal addressed my main concerns, increasing my evaluation

**Key Questions For Authors:**

1. Missing details on calibration data and sampling variability for Llama experiments. Please indicate whether the gradients employed are per sample or per batch. Also, please run ablations on the subsample size to better understand their effect on ranking stability.
2. Please report the total wall-clock time and GPU memory for compressing an entire Llama model at a given compression ratio.

**Limitations:**

Yes

**Strengths And Weaknesses:**

**Strenghts**

1. The proposed method efficiently captures off-diagonal correlations from the FIM, addressing a core limitation of diagonal Fisher approximations
2. Under stated assumptions, the authors derive GFWSVD, a closed-form solution to minimizing expected second-order loss increase.
3. Empirical results show performance gains on synthetic tasks and Transformer benchmarks.

**Weaknesses**

1. Claims rely on strong assumptions (MLE-at-optimum, MVN perturbation model, rank-1 Kronecker Fisher).
2. Approximation limited to layers, ignoring cross-layer impact.
3. As noted by the authors, rank-1 Kronecker might not be sufficient for some layers, missing curvature information.

---

> ### Author Rebuttal · Authors · 2026-03-31
>
> We appreciate that the reviewer recognized the main contribution of the paper. We address the raised weaknesses below.
>
> > strong assumptions (MLE-at-optimum, MVN perturbation model, rank-1 Kronecker Fisher).
>
> **Rank-1 Kronecker Fisher**
> As stated in Limitations, inverting a sum of Kronecker products (approximating FIM) cannot be expressed as a single Kronecker product (an open problem for the whole field), restricting weighted SVD to a single-term solution. Rank-1 Kronecker remains standard across ML tasks [1-4,6].
>
> **MVN**
> This is a local perturbation model around $W^{\star}$, not a global posterior claim. We model $\Delta W$ as MVN with covariance given by the Kronecker-factored Fisher, which matches the standard second-order approximation. It does not introduce geometric assumptions beyond local quadratic behavior, but specifies a reference distribution for measuring expected loss increase in a similar fashion to [5,6].
>
> **MLE-at-optimum**
> Standard LLM pre-training is fundamentally MLE. Applied to fully pre-trained models (BERT, Llama 2, Llama 3.1), it is well-grounded to assume weights reside near a local optimum of the log-likelihood surface. This is standard in second-order compression and quantization [7,8].
>
> > ignoring cross-layer impact
>
> This is not a conceptual gap: we follow standard layer-wise setup [1-4]. Recent theoretical findings also suggest intra-layer connections dominance [9].
> Appendix A already states that compression effectiveness varies across layers and that cross-layer coordination is not considered. We will make this explicit in the main text.
> The key contribution here is showing that a stronger within-layer Fisher model (rather than diagonal weighting) is scalable in practice, and the experiments confirm this.
>
> > rank-1 Kronecker might not be sufficient for some layers, missing curvature information.
>
> Indeed, Figure 6 and Appendix A show some layers have slowly decaying spectra, so rank-1 may miss curvature variance. This is expected and serves as a diagnostic we provide for *when the approximation is better or worse*: rank-1 Kronecker is a first scalable approximation class, not an assertion that every layer’s Fisher is exactly rank-1 Kronecker.
>
> >Missing details on calibration data and sampling variability for Llama experiments. Please indicate whether the gradients employed are per sample or per batch. Also, please run ablations on the subsample size to better understand their effect on ranking stability.
>
> We use FineWeb [10] as calibration data, collecting per-batch gradients (batch size 128) on 9600 random samples. Results for different batch sizes (LLaMA2-7B Chat, 20% compression) show negligible differences:
>
> | Setting  | WT-2 ↓ | ARC-C ↑ | ARC-E ↑ | HSwag ↑ | PIQA ↑ | WGrande ↑ | OpenBook ↑ | Avg ↑ |
> | -------- | ------ | ------- | ------- | ------- | ------ | --------- | ---------- | ----- |
> | Bs = 32  | 10.9   | 0.33    | 0.63    | 0.46    | 0.74   | 0.60      | 0.26       | 0.50  |
> | Bs = 128 | 11.1   | 0.33    | 0.62    | 0.47    | 0.74   | 0.61      | 0.25       | 0.50  |
> | Bs = 256 | 11.1   | 0.33    | 0.63    | 0.46    | 0.74   | 0.61      | 0.25       | 0.50  |
>
>
> > Please report the total wall-clock time and GPU memory for compressing an entire Llama model at a given compression ratio.
>
> Full pipeline timings (LLaMA2-7B, 4 GPUs): FWSVD/Basis Sharing ~1h, GFWSVD ~3h (incl. gradients + Fisher), single GPU: ASVD ~14h, SVD-LLM ~11h.
>
> | Method         | Time, GPU·h |
> |----------------|------------|
> | FWSVD          | 4          |
> | GFWSVD         | 12         |
> | ASVD           | 14         |
> | SVD-LLM        | 11         |
>
> Dobi-SVD/FWSVD require ~30+ hours (4 GPUs, 10 epochs finetuning).
>
> The compression script adds ~267 MB. Precomputing Kronecker factors (Alg. 1, reused across compression levels) requires ~5.5 GB, due to Lanczos procedure over calibration batches.
>
>
> **References**
>
> [1] Martens & Grosse. Optimizing neural networks with kronecker-factored approximate curvature. ICML, 2015.
>
> [2] Grosse & Martens. A kronecker-factored approximate fisher matrix for convolution layers. ICML, 2016.
>
> [3] Tang et al. SKFAC: Training neural networks with faster kronecker-factored approximate curvature. CVPR, 2021.
>
> [4] Dangel et al. Kronecker-factored approximate curvature for physics-informed neural networks. NeurIPS, 2024.
>
> [5] Weimar et al. *Fisher Information Flow in Artificial Neural Networks*. Phys.Rev. X, 15:031072, 2025.
>
> [6] Zhang et al. Noisy Natural Gradient as Variational Inference. ICML, 2018.
>
> [7] Singh & Alistarh. WoodFisher: Efficient second-order approximation for neural network pruning. NeurIPS, 2020.
>
> [8] Chee et al. QuIP: 2-bit quantization of large language models with guarantees. NeurIPS, 2024.
>
> [9] Dong et al. Towards quantifying the hessian structure of neural networks. arXiv preprint arXiv:2505.02809, 2025.
>
> [10] Penedo et al. The FineWeb Datasets: Decanting the Web for the Finest Text Data at Scale.  NuerIPS, 2024.

---

> > ### Author Rebuttal · Reviewer_oVfe · 2026-04-04
> >
> > The authors properly addressed all my comments. I have increased the score.

---

> > > ### Author Response · Authors · 2026-04-07
> > >
> > > We would like to thank the reviewer for the time and effort.

---

### Official Review · Reviewer_YwPQ · 2026-03-13

**Soundness:** 3
**Presentation:** 3
**Significance:** 2
**Originality:** 3
**Overall Recommendation:** 4
**Confidence:** 3

**Summary:**

This paper proposes Matrix-free Fisher Factorization to compress model parameters.
The method tries to balance between FWSVD and K-FAC, i.e., between simple diagonal fisher matrix and expensive computation cost, respectively.
The essential idea is to derive the optimal factorization under matrix variate normal assumpation.
Experimental results demonstrate it is effective for compressing LLama2 and LLama3.1 on WT2 dataset.

**Compliance With Llm Reviewing Policy:**

Affirmed.

**Final Justification:**

I have read rebuttal response, and I agree authors are doing a good job within their scope. But I hope more to come in the future as there are still some important topics raised in the review.

**Key Questions For Authors:**

1. how about matrix variate t distribution assumption? which is more robust than the normal assumption.
2. where does the Kronecker factors singularity mostly come from? insufficent data or model itself?
3. what is the condition number of W1 and W2 in L249?
4. what about the memory cost?
5. can Fisher matrix be dynamically updated during fine-tuning, while current version is being estimated statically after pre-training
6. have you tried multimodal models? real world applications are often multimodal, and the weight matrices for different modalities may fail your single Kronecker product approximation.

**Limitations:**

yes

**Strengths And Weaknesses:**

Strength:
1. generally applicable to different model architectures;
2. fast computation by kronecker product and lanczos decomposition;
3. good experimental results

Weakness:
1. not sure under what condition single Kronecker product is sufficient or not to approxiate the model Fisher matrix
2. unable to generalize to multiple Kronecker product, which is mathematically hard
3. possible singular decomposition A or B in Kronecker product, extra regularization cost
4. no cross layer joint compression solution
5. the theoretically optimal GFWSVD does not guarantee the best empirical performance, e.g., assumption & local optimality & layer dependency, etc., need a limitation section to discuss and summarize

---

> ### Author Rebuttal · Authors · 2026-03-31
>
> Thank you for your constructive feedback.
>
> > not sure under what condition single Kronecker product is sufficient
>
> A single Kronecker product is appropriate when the R-permuted Fisher is dominated by a leading Kronecker component. Figure 6 provides a spectral diagnostic: fast decay indicates a good rank-1 fit, while slow decay implies worse fit. Thus, this is a layer-dependent approximation whose validity can be verified empirically. We will make this criterion explicit in the main text.
>
> > unable to generalize to multiple Kronecker product
>
> There are no theoretical results allowing us to do so. Inverting a sum of Kronecker products (approximating FIM) cannot be expressed as a single Kronecker product. Thus, the problem of multi-term extension is currently open for the **whole field** and should not be attributed to our particular work. Rank-1 Kronecker approximation remains standard across various ML tasks [1–5].
>
> > possible singular A or B
>
> In LLaMA 2–7B, raw Kronecker factors are non-positive definite in \~70% of layers. We apply iterative regularization:
> $Y \rightarrow Y + \alpha Y$ , increasing $\alpha$ until Cholesky succeeds. The time overhead for this operation is negligible (\~5 seconds per layer). Regularization is also a standard practice for Fisher-based ML tasks: for example, K-FAC [1] employs Tikhonov regularization.
>
> > no cross layer solution
>
> This is an intentional scope choice following an established practice in other tasks, e.g., in optimization [1-4]. Similarly, we focus on intra-layer second-order compression and do not claim to solve cross-layer coordination. Recent theoretical findings additionally support an intra-layer connections dominance setting [6]. Appendix A already notes layer-wise variability; we will make this explicit in the main text. The contribution of our paper is to show that stronger intra-layer Fisher models remain scalable and useful.
>
> > GFWSVD optimality vs empirical performance
>
> GFWSVD is optimal for the local Fisher-weighted quadratic objective around optimal point $W^*$, not for general training dynamics. The empirical section evaluates whether this surrogate is useful; consistent improvements over diagonal baselines indicate that it is meaningful, even without global guarantees. We will make it more transparent in the main text.
>
> ## Questions
>
> > 1) Matrix-variate (t)
>
> Our derivation relies on the quadratic structure induced by the MVN model. While a matrix-variate $t$ model offers a robust heavy-tailed alternative, the Fisher-weighted SVD optimum no longer holds in closed form in this case without additional approximations. We view this as future work (robustness vs. tractability).
>
> > 2) source of singularity
>
> The best explanation is that singularity arises primarily from the mismatch between the true empirical Fisher and the rank-1 approximation, rather than from the model itself (though overparameterization may also contribute). As this is a standard discrepancy between mathematical models and real-world data, many curvature-based ML methods similarly utilize regularization (e.g., damping in K-FAC [1]).
>
> > 3) Condition numbers
> We agree this is a useful diagnostic.
> LLaMA 2–7B:
>
> - $W_1$: min 1.55, median 36.9, max 227.3
> - $W_2$: min 1.55, median 44.9, max 405.7
>
> > 4) Memory cost
>
> As our method never materializes the full Hessian, there is no significant additional memory cost; compressing LLaMA 2-7B requires only 267 MB. Precomputing Kronecker factors (Algorithm 1) via Lanczos iterations adds a one-time overhead of 5.5 GB, which is reused across all compression levels.
>
> > 5) Dynamic Fisher updates
>
> While dynamic updates are possible, our method focuses on post-training compression using a static Fisher estimate at a converged $W^{\star}$. Moving toward training-time tracking shifts the scope toward K-FAC-style optimization rather than the post-training setting studied here.
>
> > 6) Multimodal models
>
> While we have not evaluated multimodal models, our method is model-agnostic as it approximates the Hessian for individual linear layers—a fundamental building block of all Transformer architectures. We see no conceptual obstacles to applying GFWSVD to multimodal models and consider this a meaningful direction for future work.
>
> ## References
>
> [1] Martens, J., & Grosse, R. *Optimizing neural networks with Kronecker-factored approximate curvature*. ICML, 2015.
> [2] Grosse, R., & Martens, J. *A Kronecker-factored approximate Fisher matrix for convolution layers*. ICML, 2016.
> [3] Tang, Z., et al. *SKFAC: Training neural networks with faster Kronecker-factored approximate curvature*. CVPR, 2021.
> [4] Dangel, F., et al. *Kronecker-factored approximate curvature for physics-informed neural networks*. NeurIPS, 2024.
> [5] Zhang, G., et al. *Noisy natural gradient as variational inference*. ICML, 2018.Thank you for your constructive feedback.
> [6] Dong et al. *Towards quantifying the hessian structure of neural networks*. arXiv preprint arXiv:2505.02809, 2025.

---

> > ### Author Rebuttal · Reviewer_YwPQ · 2026-04-01
> >
> > I am satisfied with authors response, so I have raised my score.

---

> > > ### Author Response · Authors · 2026-04-07
> > >
> > > We would like to thank the reviewer for the time and effort.

---

### Decision · Program_Chairs · 2026-04-30

**Decision:**

Accept (regular)

**Comment:**

The paper considers post-training model compression under a parameter sensitivity framework where the model's weight matrices follow a matrix-variate normal distribution, whose covariances link to a Kronecker approximation of the Fisher.
Computing the rank-compressed weights requires a Fisher-weighted SVD, and the paper proposes a method for computing a Kronecker-factored Frobenius-optimal approximation of the empirical Fisher, which has higher accuracy than existing approaches.
The approach called 'Generalized Fisher-weighted SVC (GFWSVD)' is a generalization of previous methods, and experiments demonstrate that GFWSVD outperforms them in terms of post-pruning performance.

The paper does a good job at contrasting its contributions with existing works and provided extensive experimental evaluations, which the reviewers appreciated.
Reviewers were satisfied with the author rebuttal, which provided further details about the scope, computational cost, and experimental details.
The proposed method to compute a Frobenius-optimal Kronecker approximation of the Fisher is not novel, though, and has been proposed in the context of optimization [1] and Bayesian deep learning [2] before. These works should be added to the bibliography.
The novel contribution is the application to model compression, and
the results represent an interesting result where improved curvature approximation quality leads to better downstream performance.
Overall, I share the reviewer's assessment to accept the paper, as the proposed framework may also be applicable beyond model compression.

I would like to urge the authors to incorporate all changes from the rebuttal, as well as the following points:
- There seems to be an error in Table 3, which highlights 0.15 as largest accuracy, although there are larger numbers in the same cell (0.16, 0.17). Please fix.
- Cite the existing literature on Frobenius-optimal KFAC.
- Comment on using the empirical Fisher rather than the Fisher. This seems to be common in model pruning (e.g. [3]), and it would be good to mention that the approach can also be applied to the Fisher (gradients replaced by pseudo-gradients), and it might also be interesting to add results for the Fisher, although the field seems to rely on the empirical Fisher.

[1] Schnaus, D., Lee, J., & Triebel, R. (2021). Kronecker-factored optimal curvature. Bayesian Deep Learning NeurIPS 2021 Workshop

[2] Koroko, A., Anciaux-Sedrakian, A., Gharbia, I. B., Gares, Valerie, Haddou, M., & Tran, Q. H. (2022). Efficient approximations of the fisher matrix in neural networks using kronecker product singular value decomposition.

[3] Singh, S. P., & Alistarh, D. (2020). Woodfisher: efficient second-order approximation for neural network compression. NeurIPS.